# InEdit-Bench: Benchmarking Intermediate Logical Pathways for Intelligent Image Editing Models

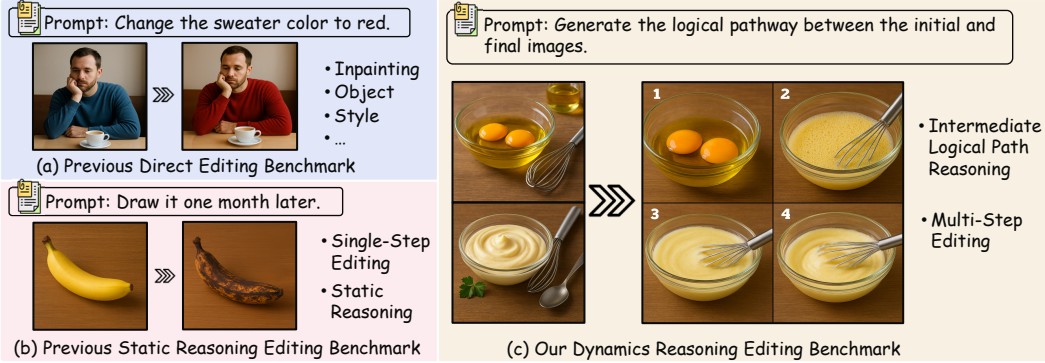

Figure 1: Comparison of previous image editing benchmarks and our proposed InEdit-Bench.

## Abstract

Multimodal generative models have made significant strides in image editing, demonstrating impressive performance on a variety of static tasks. However, their proficiency typically does not extend to complex scenarios requiring dynamic reasoning, leaving them ill-equipped to model the coherent, intermediate logical pathways that constitute a multi-step evolution from an initial state to a final one. This capacity is crucial for unlocking a deeper level of procedural and causal understanding in visual manipulation. To systematically measure this critical limitation, we introduce **InEdit-Bench**, the first evaluation benchmark dedicated to reasoning over intermediate pathways in image editing. InEdit-Bench comprises a meticulously hand-annotated dataset spanning 4 fundamental categories: state transition, dynamic process, temporal sequence, and scientific simulation, which collectively cover 16 distinct sub-tasks. We also propose a suite of 6 evaluation metrics to assess the logical coherence and visual naturalness of the generated pathways, as well as model fidelity to specified or novel path constraints. Our comprehensive evaluation of 14 representative image editing models on InEdit-Bench reveals significant and widespread shortcomings in this domain. By providing a standardized and challenging benchmark, we aim for InEdit-Bench to catalyze research and steer development towards more dynamic, reason-aware, and intelligent multimodal generative models.

## 1 Introduction

Navigating a complex task is not a single, straightforward jump from inception to conclusion. Instead, the path to the solution is comprised of a series of indispensable intermediate steps that bridge the chasm between the start and the end. Often, the true challenge lies not in the crossing itself, but in the fact that this "bridge" is not readily apparent. We can perceive the starting point and the final destination, yet the pathway connecting them remains invisible. Therefore, the ability to reconstruct this hidden path is a fundamental test of reasoning, prevalent across countless domains.

The capacity to reason about transformative pathways is of paramount importance in artificial intelligence, where generative models have unlocked unprecedented prowess in image creation Huang et al. (2024a); Pan et al. (2025a); Deng et al. (2025). Moreover, intelligent image editing, which moves beyond simple generation from scratch Fang et al. (2025); Deng et al. (2025), demands a more profound semantic understanding Wang et al. (2024); Yu et al. (2024) and precise manipulation Alaluf et al. (2021); Brooks et al. (2022), making it a crucial testbed for model competence. However, despite the remarkable achievements of leading models, they primarily focus on single-step editing and static reasoning Wu et al. (2025c); Sun et al. (2023); Deng et al. (2025), inherently lacking the ability to model process evolution. This raises a critical question at the model level: given only the starting and ending images, how can a model generate a sequence of intermediate images that adheres to causal logic and visual naturalness?

To bring greater attention to this unexplored frontier, we introduce a novel evaluation benchmark, termed **InEdit-Bench**, centered on the generation of these intermediate logical pathways. As shown in Fig. 1, our paradigm moves beyond simply appraising the final output. Instead, it challenges a model to construct the entire, coherent sequence of transformations that logically connects a given initial state to a final target. This marks a significant departure from existing benchmarks Zhao et al. (2025); Pan et al. (2025b); Wu et al. (2025d), which, while valuable for assessing static outcomes like instruction-following fidelity and semantic consistency Pan et al. (2025b); Zhang et al. (2023b), offer no quantitative measure of procedural reasoning. By shifting the evaluation focus from the "destination" to the "intermediate logical pathways", InEdit-Bench provides a more nuanced and rigorous assessment of the core reasoning faculties, such as causal understanding and strategic planning. Ultimately, our goal is to steer the research focus away from static, single-step outcomes and towards the development of models capable of true procedural and dynamic reasoning.

Our InEdit-Bench consists of 237 high-quality, meticulously hand-annotated data instances. The dataset is organized into four fundamental categories: **state transition**, **dynamic process**, **temporal sequence**, and **scientific simulation**, collectively covering 16 distinct sub-tasks. Each instance in the benchmark comprises an initial state image, a final state image, and a corresponding textual prompt. To ensure a structured output, these prompts instruct models to generate a single image divided into $N$ grids, where each grid depicts a distinct stage of the process. Furthermore, each prompt contains both a basic editing instruction and a concise summary of key intermediate stages, generated via a large multimodal model (LMM).

For a comprehensive evaluation, InEdit-Bench employs six key metrics to assess the generated procedural pathways: **appearance consistency**, **perceptual quality**, **semantic consistency**, **logical coherence**, **scientific plausibility**, and **process plausibility**. While the first three metrics are adapted from standard image editing tasks Pan et al. (2025b); Zhao et al. (2025), the latter three are novel and specifically designed for our process-oriented benchmark. These new metrics provide an objective assessment of the transition logic between stages, their scientific fidelity, and the holistic comprehension of the intermediate pathway. To automate this multifaceted evaluation, we adopt the LMM-as-a-Judge paradigm Zhao et al. (2025), where a powerful LMM serves as an objective evaluator for the generated image pathways.

Using InEdit-Bench, we conduct an evaluation of 14 representative intelligent image editing methods, including state-of-the-art models such as GPT-Image-1 OpenAI (2025), Nano-Banana Google (2025), Flux-Kontext-pro Labs (2025), Qwen-Image-Edit Wu et al. (2025a), Bagel Deng et al. (2025), Emu Sun et al. (2023), OmniGen Wu et al. (2025c), and Step1X-Edit Liu et al. (2025). Our findings consistently reveal that these models exhibit significant shortcomings in multi-step editing and dynamic reasoning, highlighting a crucial direction for future research.

In summary, our main contributions are as follows:

(1) We introduce InEdit-Bench, the first evaluation benchmark for multi-step image editing and dynamic reasoning. It provides a challenging testbed to assess the ability of a model to comprehend and generate intermediate logical pathways, catalyzing future research in controllable visual editing.

(2) We construct a high-quality, meticulously annotated dataset for the benchmark, encompassing 4 fundamental categories and 16 distinct sub-tasks. This dataset establishes a robust foundation for the systematic evaluation of complex editing capabilities.

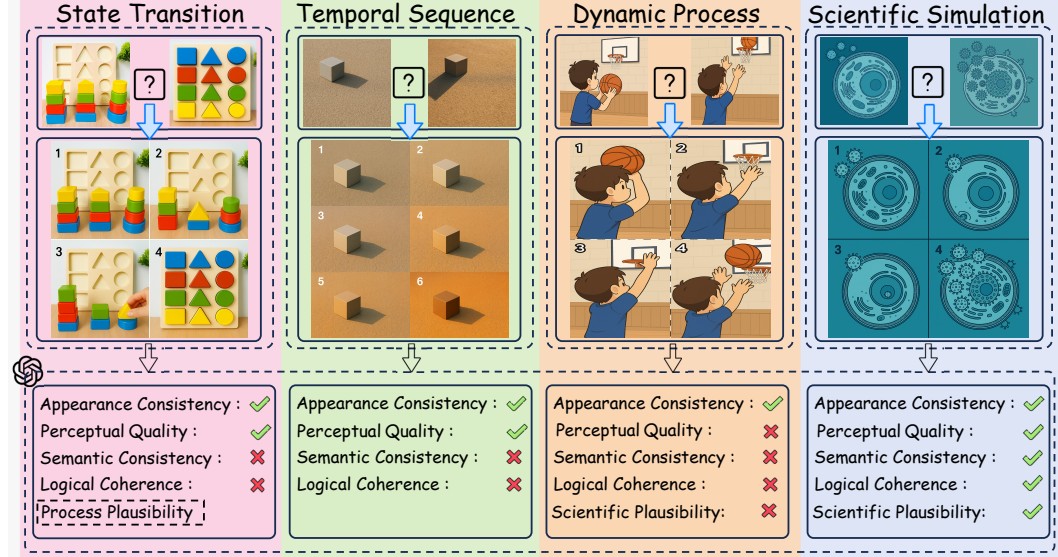

Figure 2: **Overall introduction to InEdit-Bench.** InEdit-Bench focuses on dynamic reasoning and multi-step editing modes, requiring models to generate intermediate logical pathways for given tasks. It spans 4 key domains: state transition, dynamic process, temporal sequence, and scientific simulation. The evaluation is conducted through 6 dimensions: appearance consistency, perceptual quality, semantic consistency, logical coherence, scientific plausibility, and process plausibility.

(3) We establish a multi-faceted evaluation protocol with 6 assessment dimensions. These metrics are specifically designed to capture the visual fidelity and logical coherence of intermediate paths, ensuring a rigorous and objective assessment.

(4) We present a comprehensive analysis of 14 state-of-the-art models on InEdit-Bench. Our findings reveal the significant limitations of current methods and highlight key areas for future improvement.

## 2 RELATED WORK

### 2.1 INSTRUCTION-BASED IMAGE EDITING

Instruction-based Image Editing (IIE) Wang et al. (2024); Huang et al. (2024a); Pan et al. (2025a) emphasizes the direct expression of user intent, thereby reducing interaction complexity while improving controllability and practicality. Early work such as InstructPix2Pix Brooks et al. (2022) proposed driving image editing with simple instructions, and subsequent research has continuously improved data quality and model architectures. For example, MagicBrush Zhang et al. (2023a) introduced high-quality manually annotated data, MGIE Fu et al. (2024) and SmartEdit Huang et al. (2024b) leveraged multimodal large language models to enhance semantic understanding and reasoning, while OmniGen Wu et al. (2025c) and Gemini Team & et al. (2025) built unified multimodal architectures to strengthen task generalization. Although existing methods have achieved progress in quality Podell et al. (2024); Rombach et al. (2022), efficiency Team (2024); Wu et al. (2025b), and controllability Du et al. (2025), the capabilities of models in multi-step editing and higher-order understanding remain underexplored. To address this, we propose InEdit-Bench, aiming to provide valuable insights for the community and advance the development of such models.

### 2.2 IMAGE EDITING BENCHMARKS

The proliferation of image editing benchmarks has accelerated the development of model evaluation frameworks in recent years. Benchmarks such as TedBench Kawar et al. (2023), EditVal Basu et al. (2023), and EditBench Wang et al. (2023) primarily focus on basic editing tasks, while MagicBrush Zhang et al. (2023a) provides high-quality data but its evaluation metrics Caron et al. (2021); Radford et al. (2021) have inherent limitations in fully reflecting image quality. With further research, Reason-edit Huang et al. (2024b) and RISEBench Zhao et al. (2025) have begun to em-

phasize evaluating models' understanding and reasoning capabilities under complex instructions. Complex-Edit Yang et al. (2025) introduces a chain-of-thought-like multi-step editing mechanism, while I2Ebench Ma et al. (2024) and KRIS-Bench Wu et al. (2025d) explore higher-level editing abilities from the perspectives of multi-dimensional skills and knowledge-driven reasoning. Overall, although existing benchmarks have made significant progress in terms of task scale and diversity, they remain insufficient in modeling intermediate logical pathways and supporting dynamic chain-style reasoning. To address this gap, we propose InEdit-Bench, a new benchmark for intermediate logical pathway editing, designed to systematically evaluate the capabilities of intelligent visual editing models in multi-step image editing as well as dynamic understanding and reasoning.

## 3  INEDIT-BENCH

### 3.1  BENCHMARK CONSTRUCTION

InEdit-Bench is an innovative benchmark designed to systematically evaluate a model's capability in comprehending and representing intermediate logical pathways. As illustrated in Fig. 2, we categorize editing tasks into four fundamental types based on the evolutionary dynamics of their intermediate states: **state transition**, **dynamic process**, **temporal sequence**, and **scientific simulation**. Fig. 3 details the task distribution within InEdit-Bench. The representative example images of each subtask are provided in Appx. A.6. The image sources for this benchmark include data collected from the internet under permissive licenses, images generated by generative models OpenAI (2025), and samples extracted from existing datasets Wu et al. (2025d); Lanitis et al. (2002).

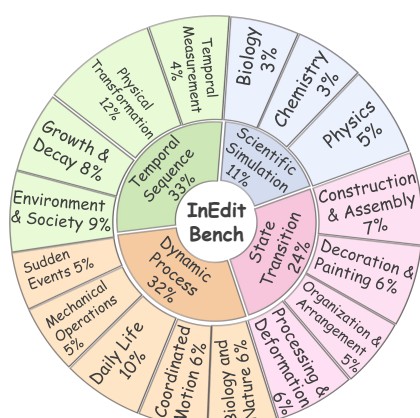

Figure 3: The task type distribution of InEdit-Bench. InEdit-Bench conducts a comprehensive evaluation of visual editing models across 16 sub-tasks under 4 domains.

### 3.1.1  STATE TRANSITION

State transition reasoning aims to understand and reconstruct the discrete changes from an initial state to a final state. The primary challenge is to identify key discrete change nodes and infer their dependencies, thereby generating a complete editing sequence that is both logically coherent and structurally clear. This task category is further subdivided into the following four subcategories. **(1) Construction and Assembly:** This subtask requires the model to combine multiple independent components into a complete entity based on spatial logic. The main difficulty lies in structural reconstruction and resolving the dependencies between assembly steps. **(2) Decoration and Painting:** This pertains to applying colors, textures, or patterns to a target's surface. The critical aspect is achieving precise region identification and control over the sequence of operations. **(3) Organization and Arrangement:** The central difficulty here is the systematic organization and spatial arrangement of elements, which requires the model to possess strong reasoning capabilities regarding layout structures and spatial relationships. **(4) Processing and Deformation:** This involves the model understanding how an operation alters the physical properties or state of an object. The key challenge is to precisely capture and simulate the object's morphological transformations.

### 3.1.2  DYNAMIC PROCESS

Dynamic process reasoning is characterized by a continuous transformation from an initial state to a final one. In contrast to scenarios involving discrete state transition, this paradigm challenges the model to process seamless and continuous progressions, demanding that every intermediate step demonstrates both natural fluidity and logical consistency. The tasks within this category are further classified into five distinct subdomains. **(1) Biology and Nature:** Focuses on the evolution of organisms and natural phenomena, stressing deduction from biological characteristics and the laws of nature (*e.g.*, a spider constructing a web, a chick hatching). **(2) Coordinated Motion:** Concerns the fluid and coordinated movement of entities through space, mandating that the model comprehend

motion dynamics to produce smooth, logically connected intermediate actions (*e.g.*, a long jump, a basketball shot). **(3) Daily Life:** Encompasses the modeling of common operations, interactions, and behaviors observed in everyday contexts. **(4) Mechanical Operation:** Involves illustrating the continuous structural alteration of an object via incremental reasoning, highlighting the operational mechanisms that propel such changes (*e.g.*, a compressor flattening a cube). **(5) Sudden Events:** Characterized by abrupt, often destructive transformations, requiring the model to identify pivotal moments and render believable visual consequences (*e.g.*, a building demolition). To systematically guide the model in generating these continuous transformations, the task instructions for each instance provide one to three key path-node prompts.

### 3.1.3 TEMPORAL SEQUENCE

Temporal sequence reasoning is concerned with the progressive evolution of a target state over time. Diverging from the emphasis on continuity in dynamic process, temporal tasks prioritize identifying and demarcating critical change points on a timeline. Such tasks necessitate that the model is endowed with a capacity for temporal-aware inference and can accurately represent the distinct characteristics of each evolutionary phase. The editing protocol requires the model to partition the entire process into uniform temporal intervals, thus providing a clear and complete trajectory of the temporal evolution. This domain is subdivided into the following four subcategories. **(1) Environment and Society:** Concerns the progressive evolution of environmental phenomena and social behaviors (*e.g.*, a train arriving at a station, the formation of sand ripples), requiring the model to reason about and generate temporally congruent sequences for such events. **(2) Growth and Decay:** Pertains to the life cycles of organisms (*e.g.*, a flower blooming, a wound healing), requiring the generation of a time-series that aligns with biological principles. **(3) Physical Transformation:** Typically involves alterations in the physical properties of materials or objects (*e.g.*, ice melting). The primary difficulty lies in modeling the temporal progression of these properties while ensuring the logical coherence of the transitional states. **(4) Temporal Measurement:** Focuses on the representation of time as a quantifiable metric (*e.g.*, a progress bar, an hourglass), demanding precise reasoning about quantitative changes along the temporal dimension.

### 3.1.4 SCIENTIFIC SIMULATION

Scientific simulation is designed to model principles from the fields of **Physics**, **Chemistry**, and **Biology**. These domains impose a strict requirement on the model to adhere to scientific laws, while illuminating the intermediate logical steps of a given process. For instance, physical phenomena range from the diffusion effect to total internal reflection; chemical processes include reactions such as the combustion of a magnesium strip and displacement reactions; and biological principles are exemplified by life processes like cell division and DNA replication. These tasks require the model to comprehend and execute complex scientific procedures, deduce causal mechanisms, and render each pivotal stage. To streamline this process, task instructions provide concise keyframe prompts, ensuring the model concentrates on the most critical phases and disregards superfluous steps.

## 3.2 EVALUATION METRICS

Diverging from conventional benchmarks that assess single-step image editing, InEdit-Bench fundamentally redefines the evaluation paradigm by reorienting the focus from input-output comparisons to the procedural integrity of the entire transformation. To this end, we develop a multi-faceted evaluation framework built upon six key dimensions. These are categorized into two groups: three foundational metrics for visual quality (**Appearance Consistency**, **Perceptual Quality**, **Semantic Consistency**), and three novel dimensions (see Fig. 4) designed to scrutinize the plausibility of intermediate processes (**Logical Coherence**, **Scientific Plausibility**, **Process Plausibility**). Our evaluation employs the LMM-as-a-Judge methodology, leveraging GPT-4o for its advanced reasoning capabilities, which are essential for our novel process-oriented metrics. For evaluation, the model receives the user instruction, a scoring rubric, and the generated output—a single image depicting the process across a grid, based on which it provides a numerical score for each dimension.

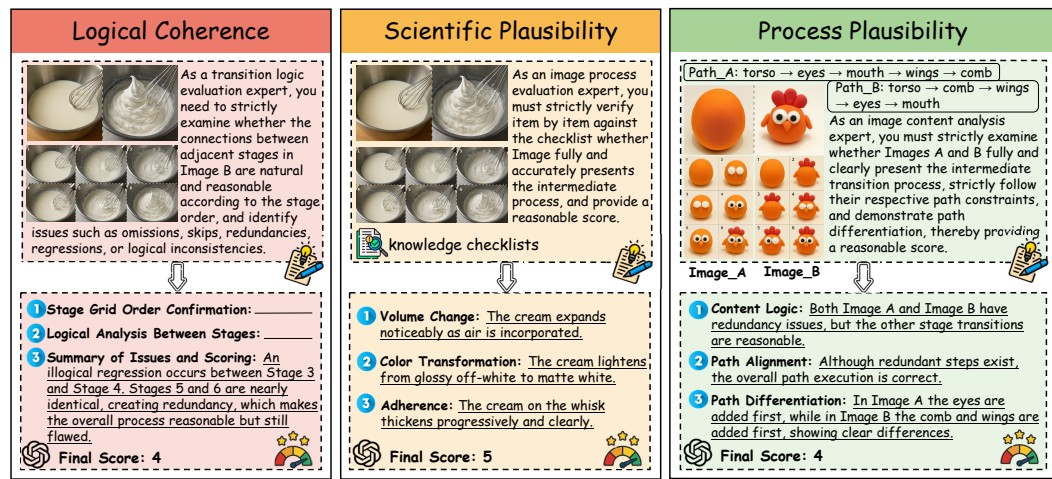

Figure 4: The evaluation metrics of **Logical Coherence**, **Scientific Plausibility**, and **Process Plausibility** in InEdit-Bench.

## 3.3 STANDARD VISUAL QUALITY METRICS

To establish a baseline for visual quality, our framework incorporates three foundational metrics from the image editing domain. Appearance consistency assesses the preservation of style and visual attributes across all depicted stages of the process. Perceptual quality measures the realism and fidelity of the generated imagery, ensuring it is free from artifacts. Semantic consistency evaluates the alignment of the final image content with the specified editing objective. Collectively, these metrics provide a robust assessment of the visual integrity of the final output.

A critical guideline for our benchmark stems from the use of a grid-based representation. The evaluation protocol explicitly requires that visual artifacts introduced by the grid format (such as grid lines, segmentation effects, and numbering) must be disregarded during content assessment. Furthermore, any layout or compositional discrepancies arising as a direct consequence of the grid structure are also to be excluded from the evaluation.

## 3.4 PROPOSED PROCESS-ORIENTED METRICS

**Logical Coherence.** Logical coherence is paramount in evaluating multi-step image editing, as it examines the integrity of the generated process in terms of its logical progression and natural flow. The assessment protocol begins by establishing the sequence of the depicted stages, applying a top-to-bottom, left-to-right convention whenever the intended order is not visually apparent. The core of the evaluation then involves a close examination of the transitions between adjacent stages for logical soundness and naturalness. This scrutiny ensures that the overall evolution is fluid and coherent, devoid of any jarring discontinuities or superfluous, repetitive actions.

**Scientific Plausibility.** Drawing inspiration from KRIS-Bench Wu et al. (2025d) and WorldGen-Bench Zhang et al. (2025), we incorporate scientific plausibility as a dedicated evaluation dimension. This metric is applied to tasks involving dynamic process and scientific simulation, where adherence to scientific logic is assessed via a series of knowledge checklists. These checklists meticulously annotate the critical features and inherent mechanisms that should be present in the intermediate stages. The assessment conducts a direct comparison of the generated visual content against the checklist's items to verify compliance with the predefined scientific standards.

**Process Plausibility.** For a comprehensive evaluation of how well a model understands intermediate pathways, we employ two prompting schemes with distinct path constraints, applied to a subset of our state transition tasks. This approach is motivated by the fact that many real-world processes are non-deterministic, often presenting multiple viable routes to the same outcome. Consequently, a capable model must not only grasp the fundamental operation of each step but also discern a rational sequence from among multiple viable paths, all while maintaining both consistency and accuracy.

Table 1: **Performance of various models on InEdit-Bench.** For the dynamic process and scientific simulation tasks, scores in gray denote performance calculated without the *scientific plausibility* metric. For the state transition task, scores in gray denote the model's performance on *process plausibility* data. The best and second-best results are highlighted in **bold** and with an underline, respectively.

| Models | State Transition | Temporal Sequence | Dynamic Process | Scientific Simulation | Overall Average | Accuracy |
|---|---|---|---|---|---|---|
| **Proprietary Models** | | | | | | |
| GPT-Image-1 | **81.12 [89.00]** | **81.25** | **79.85** (82.21) | **82.61** (84.78) | **81.33** | **16.75%** |
| Nano-Banana | 70.15 [75.70] | 74.24 | 77.85 (79.33) | 79.57 (79.89) | 75.23 | 13.30% |
| Flux-Kontext-pro | 51.76 [47.10] | 52.18 | 56.23 (59.33) | 51.30 (53.26) | 51.46 | 0.99% |
| Doubao-SeedEdit-3.0 | 38.36 [25.00] | 42.23 | 39.54 (39.81) | 33.48 (35.60) | 36.50 | 0.00% |
| **Open-Source Models** | | | | | | |
| Qwen-Image-Edit | **44.77 [51.50]** | **52.08** | **52.92** (54.23) | **44.13** (45.92) | **49.60** | 0.49% |
| Emu1 | 12.63 [3.70] | 14.96 | 14.54 (17.02) | 12.39 (13.86) | 11.42 | 0.00% |
| Emu2 | 33.46 [15.40] | 34.55 | 34.08 (37.31) | 31.30 (32.61) | 29.61 | 0.00% |
| Bagel | 38.27 [42.60] | 42.61 | 42.31 (44.33) | 39.13 (40.49) | 40.70 | 0.00% |
| Bagel-Think | 41.84 [26.50] | 44.79 | 46.92 (49.81) | 43.48 (46.74) | 40.70 | **0.99%** |
| OmniGen | 10.46 [14.00] | 16.29 | 16.15 (16.54) | 12.39 (12.50) | 14.34 | 0.00% |
| OmniGen2 | 38.39 [30.90] | 43.47 | 41.85 (44.42) | 34.35 (37.23) | 37.92 | 0.49% |
| Step1X-Edit(v1.0) | 17.18 [9.60] | 17.23 | 20.99 (22.11) | 13.91 (14.67) | 16.43 | 0.00% |
| Step1X-Edit(v1.1) | 23.47 [26.50] | 32.20 | 37.54 (39.13) | 33.91 (33.97) | 31.39 | 0.00% |
| InstructPix2Pix | 28.57 [0.00] | 37.07 | 26.38 (29.33) | 24.35 (27.99) | 23.23 | 0.00% |

Accordingly, we introduce process plausibility as an advanced metric that evaluates a model's comprehension of intermediate logical pathways. Under this metric, a successful generation must adhere to the global path sequence and avoid internal logical errors or representational deviations, thereby ensuring a clear and accurate depiction of the intended logical trajectory. A comparative analysis of model performance under the two constraint schemes reveals the overall grasp of path constraints and the capability to follow a specified path.

# 4 EXPERIMENTS

## 4.1 EXPERIMENTS SETUP

On InEdit-Bench, we evaluated 14 representative visual editing models, conducting a detailed quantitative analysis of their performance across six dimensions. The evaluated models include proprietary models: GPT-Image-1 OpenAI (2025), Nano-Banana Google (2025), Flux-Kontext-pro Labs (2025), and Doubao-SeedEdit-3.0-i2i ByteDance (2025); as well as open-source models: Qwen-Image-Edit Wu et al. (2025a), Bagel Deng et al. (2025), OmniGen Xiao et al. (2025), OmniGen2 Wu et al. (2025c), Step1X-Edit Liu et al. (2025), Emu1 Sun et al. (2024), Emu2 Sun et al. (2023), and InstructPix2Pix Brooks et al. (2022). These models cover a range of mainstream generative architectures, including autoregressive generation paradigms Sun et al. (2023); Deng et al. (2025), diffusion model architectures Brooks et al. (2022), and diffusion transformer architectures Liu et al. (2025). All generation and evaluation processes are conducted on L20 GPUs using default hyperparameter settings to ensure fairness and reproducibility. Additionally, since some open-source models do not support multi-image input, we uniformly concatenated the initial and final state images into a single image, with the initial state image placed on the left or top, and the final state image on the right or bottom, separated by a black-and-white striped line.

## 4.2 RESULT ANALYSIS

### 4.2.1 RESULTS ANALYSIS BY TASKS

Tab. 1 reports the scores of 14 models on four fundamental tasks. All scores are normalized to a 100-point scale and are evaluated by GPT-4o-2024-11-20. Results on InEdit-Bench show that GPT-Image-1 is the best-performing proprietary model, with a score of 81.33 and an accuracy of 16.75%. Among open-source models, Qwen-Image-Edit and Bagel-Think perform relatively well, scoring

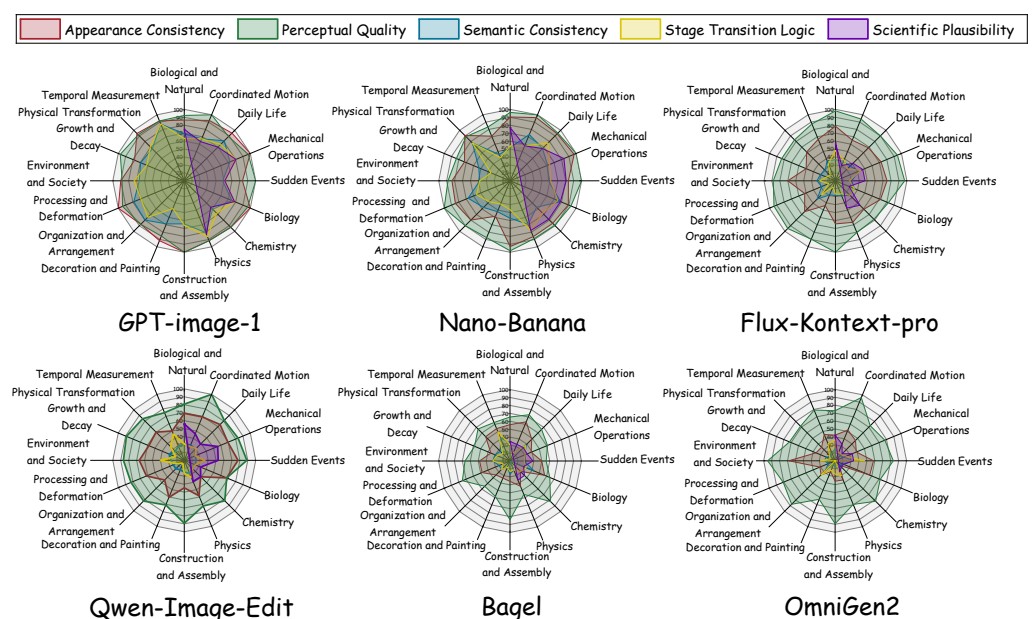

Figure 5: The performance of GPT-Image-1, Nano-Banana, Flux-Kontext-pro, Qwen-Image-Edit, Bagel and OmniGen2 across 16 sub-tasks.

49.60 and 40.70 points, respectively. Although there is still a gap between open-source and proprietary models, some open-source models have nevertheless demonstrated outstanding capabilities.

From the perspective of task dimensions, state transition task poses significant challenges for models. The average scores of all models in this category are lower than their performance on temporal sequence and dynamic process tasks. Notably, nine models, including GPT-Image-1 and Nano-Banana, score lower on state transition task than on scientific simulation task (when comparing state transition, dynamic process, and scientific simulation tasks, only the average scores of four metrics—appearance consistency, perceptual quality, semantic consistency, and logical coherence—are uniformly calculated). Furthermore, apart from the generally stronger GPT-Image-1 and Nano-Banana, all other models achieve lower average scores on scientific simulation task compared to their performance on dynamic process task. These phenomena reveal a layered structure of task complexity: from continuous to discrete, and from surface-level phenomena to deeper scientific principles, model performance shows a step-by-step decline. This highlights the limitations of current large models in complex logical reasoning and scientific law modeling.

In terms of process plausibility, GPT-Image-1 achieves the highest score of 89.00, demonstrating its advantage in understanding and articulating reasoning paths, with Nano-Banana ranking closely behind. In contrast, most of the other models generally struggle to fully meet task requirements. With respect to accuracy, even the best-performing model, GPT-Image-1, attains only 16.75%, followed by Nano-Banana at 13.30%. The remaining models achieve accuracies below 1.00%, with only Flux-Kontext-pro, Qwen-Image-Edit, Bagel-Think, and OmniGen2 reaching 0.99%, 0.49%, 0.99%, and 0.49%, respectively. Most other models yield 0% accuracy. Overall, these results reveal that current models still face significant limitations in long-term dependency capture and multi-stage causal reasoning. Achieving accurate modeling and representation of intermediate reasoning paths remains a key challenge that urgently needs to be addressed in the future.

Fig. 5 presents the performance of several representative models on 16 subtasks, while the complete results for the remaining models are provided in Appx. A.2. Overall, GPT-Image-1 demonstrates stable and superior performance across all subtasks. In contrast, the results of Flux-Kontext-pro, Qwen-Image-Edit, Bagel, and OmniGen2 exhibit significant fluctuations, particularly in the dimensions of semantic consistency and logical coherence, where substantial performance gaps may arise even among subtasks within the same basic task category. Specifically, within the temporal sequence category, Flux-Kontext-pro and Bagel experience a notable drop in performance on the physical transformation subtask, while Qwen-Image-Edit and OmniGen2 show marked degradation on the

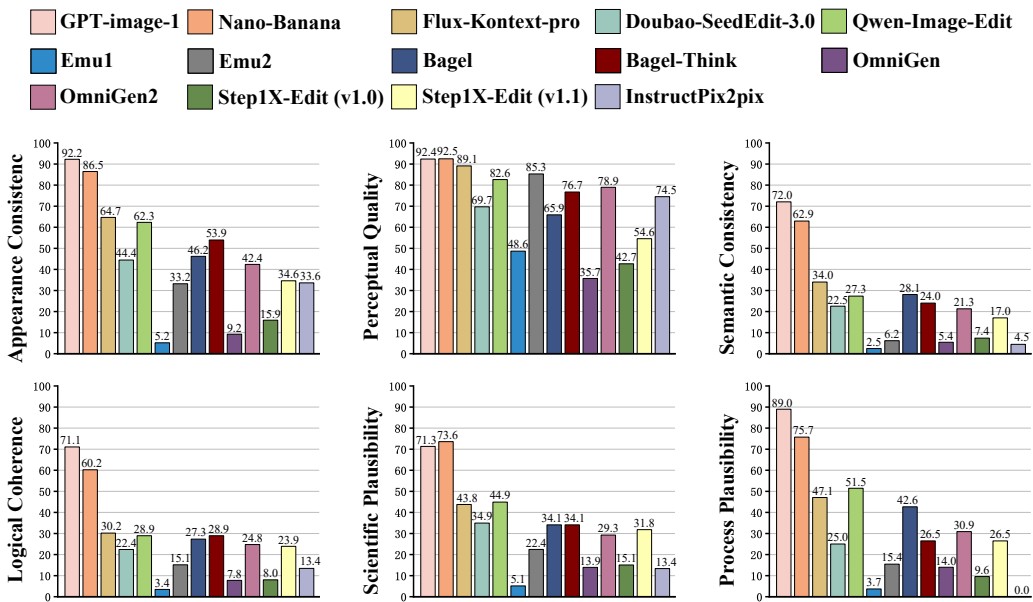

Figure 6: Comparison of models across six evaluation dimensions.

growth and decay subtask. Nano-Banana achieves relatively better average performance on dynamic process and scientific simulation tasks. However, it still suffers from performance decline and uneven results in the state transition and temporal sequence categories.

### 4.2.2 RESULTS ANALYSIS BY METRICS

As illustrated in Fig. 6, our analysis indicates that proprietary models, specifically GPT-Image-1 and Nano-Banana, consistently lead in appearance consistency, semantic consistency, and logical coherence, demonstrating their robust and well-rounded capabilities. While open-source models post lower aggregate scores, several demonstrate significant potential in specific dimensions. For instance, Qwen-Image-Edit stands out among open-source solutions for the high performance in semantic consistency, logical coherence, and scientific plausibility, occasionally rivaling proprietary counterparts. Similarly, the Bagel series is highly competitive in perceptual quality and semantic consistency, a strength also exhibited by OmniGen2 in perceptual quality.

On the other hand, open-source models Emu1 and OmniGen struggle to maintain effective visual consistency. Among proprietary models, Doubao lags behind, with significantly lower scores in appearance consistency, semantic consistency, and logical coherence compared with its peers. This suggests that Doubao may place more emphasis on rapid local editing while lacking robustness in modeling global consistency and cross-modal logical constraints. Notably, half of the open-source models score below 10.00 in the semantic consistency dimension, further underscoring their systematic deficiencies in intermediate logical path editing, instruction understanding, and semantically effective editing.

## 5 CONCLUSION

In this paper, we propose InEdit-Bench, the first system evaluation benchmark focused on image multi-step editing and intermediate logical path reasoning. It covers four basic categories: state transition, dynamic process, temporal sequence, and scientific simulation, with 16 sub-tasks. Based on this, we design six evaluation dimensions: appearance consistency, perceptual quality, semantic consistency, logical coherence, scientific plausibility, and process plausibility, to comprehensively measure the ability of intelligent visual editing models in intermediate logical path reasoning and expression. Through the evaluation of 14 representative models, we reveal significant shortcomings in current models regarding multi-step editing and dynamic reasoning capabilities, providing clear directions and reference for further optimization of model performance.

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

# A APPENDIX

## A.1 OVERVIEW OF THE APPENDIX

This appendix supplements the proposed InEdit-Bench with details excluded from the main paper due to space constraints.

The appendix is organized as follows:

- Sec. A.2: More Detailed Evaluation Results.
- Sec. A.3: Data Source of InEdit-Bench.
- Sec. A.4: Limitations.
- Sec. A.5: Utilization of Large Language Models.
- Sec. A.6: Representative Examples of InEdit-Bench Subtasks.
- Sec. A.7: Detailed Outputs of Evaluated Models.
- Sec. A.8: Design of the Prompt.

## A.2 MORE DETAILED EVALUATION RESULTS

In this section, we present a more detailed evaluation of the models, offering further analysis of their capabilities. This includes:

(1) The specific scores of 14 models across four fundamental task categories and 16 sub-tasks.

(2) The accuracy of these 14 models across the 16 sub-tasks.

### A.2.1 SCORES OF MODELS ACROSS 4 TASKS AND 16 SUB-TASKS

Tab. 2 and Fig. 7 show the specific scores of 14 models across four fundamental tasks and 16 sub-tasks, respectively. Compared to open-source models, proprietary models exhibit significantly more balanced performance. In all tasks, the models particularly excel in the perceptual quality assessment dimension, demonstrating their ability to generate natural, smooth, and high-quality images, avoiding common issues such as distortion and blurring. This result indicates that current models effectively address challenges related to image quality when generating visual content.

Most models score slightly lower in the appearance consistency dimension compared to the perceptual quality dimension, yet still demonstrate considerable capability. However, some models have still failed to effectively adapt to the new paradigm of intermediate logic path editing, resulting in poor performance in the appearance consistency dimension. For example, models like Emu1 and Omnigen encounter significant obstacles in this dimension, with performance far below that of other models.

There are significant variations in performance across models in terms of semantic consistency and logical consistency. Except for GPT-Image-1 and Nano-Banana, other models show significant imbalances in these two dimensions, with a noticeable drop in scores. Notably, models like Emu1 and Omnigen score almost zero in both semantic consistency and logical consistency, highlighting the limitations of current models in handling complex logical relationships.

Among the open-source models, all except Qwen-Image-Edit, Bagel, and Bagel-Think show relatively poor performance. Among them, a few models, such as Omnigen2 and Step1X-Edit(v1.1), show slight improvements in certain sub-tasks, achieving some scores. However, these advancements are not applicable to a broader range of tasks. Overall, the performance of open-source models still lags significantly behind that of proprietary models, with notable gaps in multiple key dimensions.

### A.2.2 ACCURACY OF MODELS ACROSS 16 SUB-TASKS

Tab. 3 shows the accuracy scores of each model across four basic categories and 16 sub-tasks. The effective scores are primarily concentrated in GPT-Image-1 and Nano-Banana. Both models perform well in multiple sub-tasks. Although GPT-Image-1 outperforms Nano-Banana overall, Nano-

Table 2: The specific scores of the models across four fundamental tasks, with metrics including Appearance Consistency (AC), Perceptual Quality (PQ), Semantic Consistency (SC), Logical Coherence (LC), Scientific Plausibility (SP). The performance of open-source and proprietary models is separately marked with the best performance in **bold**, and the second best underlined.

| | Metric | Proprietary Models | | | | Open-Source Models | | | | | | | | | |
| --- | --- | --- | --- | --- | --- | --- | --- | --- | --- | --- | --- | --- | --- | --- | --- |
| | | GPT-Image-1 | Nano-Banana | Flux-Kontext-pro | Doubao-SeedEdit-3.0 | Qwen-Image-Edit | Emu1 | Emu2 | Bagel | Bagel-Think | OmniGen | OmniGen2 | Step1X-Edit (v1.0) | Step1X-Edit (v1.1) | InstructPix2Pix |
| State Transition | AC | **95.4** | 79.6 | 56.1 | 42.7 | **53.1** | 3.1 | 30.1 | 38.3 | 48.5 | 5.6 | 31.6 | 12.8 | 21.9 | 24.5 |
| | PQ | 92.3 | 94.4 | **94.8** | 69.4 | 81.6 | 44.9 | **83.3** | 69.4 | 78.1 | 29.1 | 80.1 | 42.7 | 47.4 | 73.0 |
| | SC | **72.4** | 58.7 | 31.6 | 21.9 | **24.5** | 1.5 | 8.7 | 24.0 | 19.9 | 2.0 | 19.4 | 7.7 | 10.7 | 4.6 |
| | LC | **64.3** | 48.0 | 24.5 | 19.4 | 19.9 | 1.0 | 11.7 | 21.4 | 20.9 | 5.1 | **22.4** | 5.6 | 13.8 | 12.2 |
| | Avg | **81.1** | 70.2 | 51.8 | 38.4 | **44.8** | 12.6 | 33.5 | 38.3 | 41.8 | 10.5 | 38.4 | 17.2 | 23.5 | 28.6 |
| Temporal Sequence | AC | **89.4** | 85.2 | 64.4 | 49.2 | **62.1** | 6.8 | 35.0 | 47.7 | 55.7 | 9.5 | 47.7 | 16.3 | 35.2 | 45.1 |
| | PQ | **89.8** | 87.1 | 83.7 | 69.3 | 83.3 | 45.5 | **87.3** | 64.4 | 75.0 | 40.2 | 79.5 | 37.5 | 50.8 | 78.5 |
| | SC | **71.6** | 64.4 | 33.7 | 23.1 | 28.4 | 2.7 | 1.5 | **29.2** | 20.8 | 6.1 | 21.2 | 6.8 | 16.7 | 5.4 |
| | LC | **74.2** | 60.2 | 26.9 | 27.3 | **34.5** | 4.9 | 14.4 | 29.2 | 27.7 | 9.5 | 25.4 | 8.3 | 26.1 | 19.3 |
| | Avg | **81.3** | 74.2 | 52.2 | 42.2 | **52.1** | 15.0 | 34.6 | 42.6 | 44.8 | 16.3 | 43.5 | 17.2 | 32.2 | 37.1 |
| Dynamic Process | AC | 91.9 | **92.7** | 70.8 | 44.6 | **71.9** | 5.0 | 34.6 | 51.5 | 58.1 | 12.3 | 46.9 | 20.4 | 41.9 | 30.4 |
| | PQ | 94.6 | **97.3** | 93.5 | 70.0 | 85.8 | 55.0 | **88.5** | 64.6 | 76.2 | 36.9 | 79.2 | 48.0 | 65.4 | 71.5 |
| | SC | **71.9** | 62.7 | 36.2 | 23.1 | 28.8 | 3.8 | 8.1 | **31.2** | 29.6 | 7.7 | 23.1 | 9.2 | 20.4 | 3.8 |
| | LC | **70.4** | 64.6 | 36.9 | 21.5 | 30.4 | 4.2 | 18.1 | 30.0 | **35.4** | 9.2 | 28.5 | 10.8 | 28.8 | 11.5 |
| | SP | 70.4 | **71.9** | 43.8 | 38.5 | **47.7** | 4.6 | 21.2 | 34.2 | 35.4 | 14.6 | 31.5 | 16.5 | 31.2 | 14.6 |
| | Avg | **79.8** | 77.8 | 56.2 | 39.5 | **52.9** | 14.5 | 34.1 | 42.3 | 46.9 | 16.2 | 41.8 | 21.0 | 37.5 | 26.4 |
| Scientific Simulation | AC | **94.6** | 87.0 | 66.3 | 33.7 | **55.4** | 5.4 | 30.4 | 43.5 | 48.9 | 7.6 | 37.0 | 8.7 | 39.1 | 29.3 |
| | PQ | **93.5** | 90.2 | 80.4 | 70.7 | 73.9 | 47.8 | 75.0 | 66.3 | **80.4** | 33.7 | 73.9 | 42.4 | 50.0 | 75.0 |
| | SC | **72.8** | 68.5 | 33.7 | 20.7 | **26.1** | 0.0 | 8.7 | 25.0 | **26.1** | 4.3 | 20.7 | 3.3 | 21.7 | 3.3 |
| | LC | **78.3** | 73.9 | 32.6 | 17.4 | 28.3 | 2.2 | 16.3 | 27.2 | **31.5** | 4.3 | 17.4 | 4.3 | 25.0 | 4.3 |
| | SP | 73.9 | **78.3** | 43.5 | 25.0 | **37.0** | 6.5 | 26.1 | 33.7 | 30.4 | 12.0 | 22.8 | 10.9 | 33.7 | 9.8 |
| | Avg | **82.6** | 79.6 | 51.3 | 33.5 | **44.1** | 12.4 | 31.3 | 39.1 | 43.5 | 12.4 | 34.3 | 13.9 | 33.9 | 24.3 |

Banana still has an advantage in certain sub-tasks. In the case of open-source models, all models have an accuracy of 0% in the state transition and scientific simulation category tasks, and in tasks from other categories, only a few models show slight improvements in their scores. Overall, even the most advanced models achieve only 16.75% accuracy, with more than half of the models scoring 0%. This indicates that current models are still in the early stages of solving intermediate logic path editing tasks, far from meeting the requirements for practical application.

## A.3    DATA SOURCE OF INEDIT-BENCH

Input images for the InEdit-Bench dataset are primarily sourced from the following categories:

(1) Images generated by image generation models.

(2) Images derived from existing datasets and benchmarks.

(3) Images collected from the internet under permissive licenses.

## A.4    LIMITATIONS

This study aims to establish a pioneering benchmark for intermediate logical reasoning and multi-step editing tasks. However, as an initial exploration, the current benchmark still has several aspects that require improvement. We openly acknowledge its potential limitations, such as the dataset's insufficient scale to cover all complex scenarios and the task categorization that may not exhaust all possibilities. Future work will focus on addressing these issues to build a more comprehensive and robust benchmark.

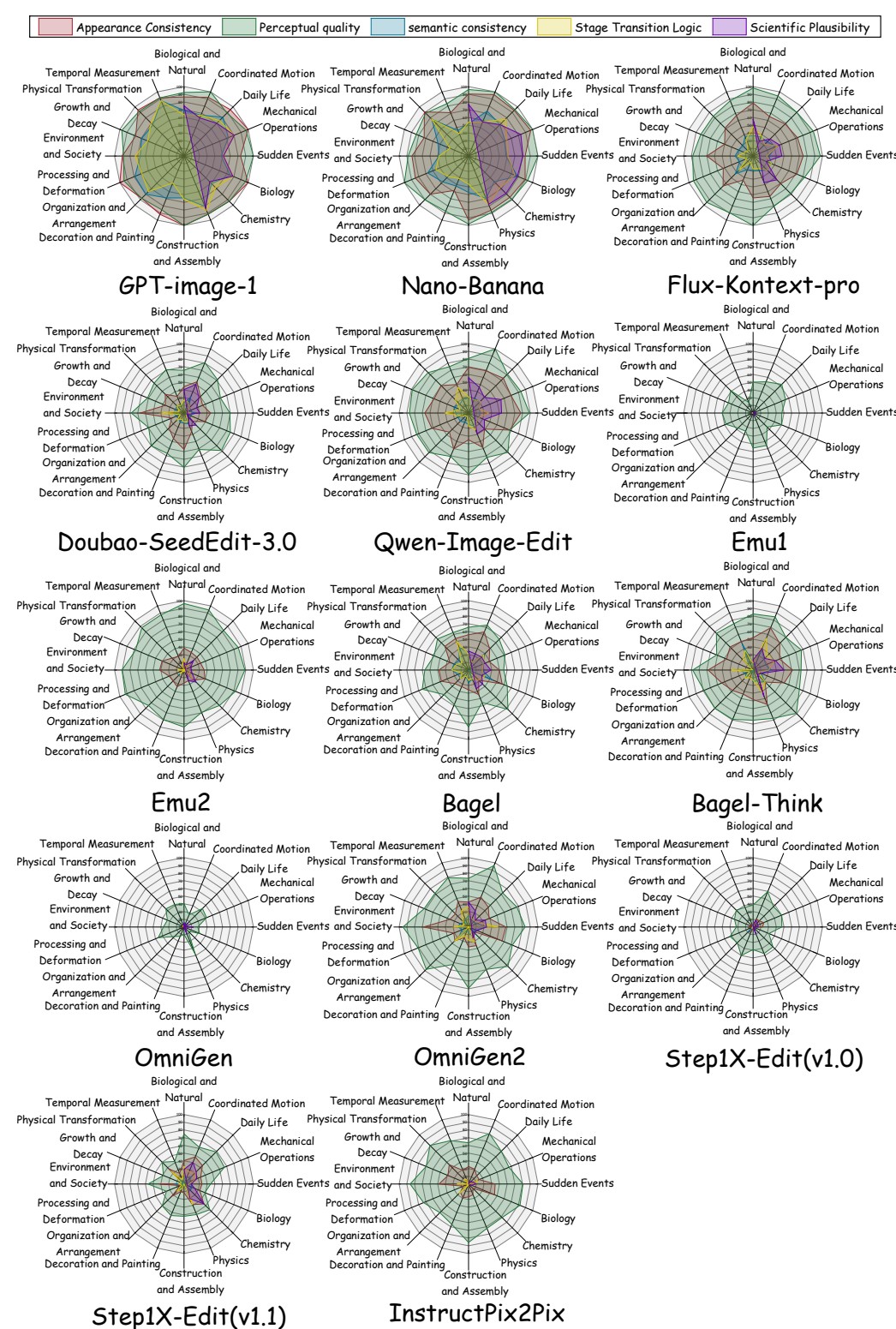

Figure 7: The average scores for 14 models across 16 sub-tasks.

Table 3: Accuracy performance of different models across 16 sub-tasks, including State Transition: Construction and Assembly (CA), Decoration and Painting (DP), Organization and Arrangement (OA), Processing and Deformation (PD). Temporal Sequence: Environment and Society (ES), Growth and Decay (GD), Physical Transformation (PT), Temporal Measurement (TM). Dynamic Process: Biology and Nature (BN), Coordinated Motion (CM), Daily Life (DL), Mechanical Operations (MO), Sudden Events (SE). Scientific Simulation: Biology (BI), Chemistry (CH), Physics (PH). The performance of open-source and proprietary models is separately marked, with the best performance in **bold** and the second-best performance underlined.

| SubTasks | | Proprietary Models | | | | Open-Source Models | | | | | | | | | |
|---|---|---|---|---|---|---|---|---|---|---|---|---|---|---|---|
| | | GPT-Image-1 | Nano-Banana | Flux-Kontext-pro | Doubao-SeedEdit-3.0 | Qwen-Image-Edit | Emu1 | Emu2 | Bagel | Bagel-Think | OmniGen | OmniGen2 | Step1X-Edit (v1.0) | Step1X-Edit (v1.1) | InstructPix2Pix |
| *State Transition* | CA | **14.29** | 7.14 | 0.00 | 0.00 | 0.00 | 0.00 | 0.00 | 0.00 | 0.00 | 0.00 | 0.00 | 0.00 | 0.00 | 0.00 |
| | DP | **8.33** | 8.33 | 0.00 | 0.00 | 0.00 | 0.00 | 0.00 | 0.00 | 0.00 | 0.00 | 0.00 | 0.00 | 0.00 | 0.00 |
| | OA | **40.00** | 10.00 | 10.00 | 0.00 | 0.00 | 0.00 | 0.00 | 0.00 | 0.00 | 0.00 | 0.00 | 0.00 | 0.00 | 0.00 |
| | PD | 7.69 | 15.38 | 0.00 | 0.00 | 0.00 | 0.00 | 0.00 | 0.00 | 0.00 | 0.00 | 0.00 | 0.00 | 0.00 | 0.00 |
| | Avg | **16.33** | 10.20 | 2.04 | 0.00 | 0.00 | 0.00 | 0.00 | 0.00 | 0.00 | 0.00 | 0.00 | 0.00 | 0.00 | 0.00 |
| *Temporal Sequence* | ES | 10.53 | 15.79 | 0.00 | 0.00 | 0.00 | 0.00 | 0.00 | 0.00 | 0.00 | 0.00 | 5.26 | 0.00 | 0.00 | 0.00 |
| | GD | **20.00** | 0.00 | 0.00 | 0.00 | 0.00 | 0.00 | 0.00 | 0.00 | 0.00 | 0.00 | 0.00 | 0.00 | 0.00 | 0.00 |
| | PT | 12.00 | 40.00 | 0.00 | 0.00 | 0.00 | 0.00 | 0.00 | 0.00 | 0.00 | 0.00 | 0.00 | 0.00 | 0.00 | 0.00 |
| | TM | **57.14** | 0.00 | 0.00 | 0.00 | 0.00 | 0.00 | 0.00 | 0.00 | 14.29 | 0.00 | 0.00 | 0.00 | 0.00 | 0.00 |
| | Avg | 18.18 | **19.70** | 0.00 | 0.00 | 0.00 | 0.00 | 0.00 | 0.00 | 1.52 | 0.00 | 1.52 | 0.00 | 0.00 | 0.00 |
| *Dynamic Process* | BN | **15.38** | 7.69 | 0.00 | 0.00 | 0.00 | 0.00 | 0.00 | 0.00 | 0.00 | 0.00 | 0.00 | 0.00 | 0.00 | 0.00 |
| | CM | 0.00 | **7.69** | 0.00 | 0.00 | 7.69 | 0.00 | 0.00 | 0.00 | 7.69 | 0.00 | 0.00 | 0.00 | 0.00 | 0.00 |
| | DL | **28.57** | 9.52 | 4.76 | 0.00 | 0.00 | 0.00 | 0.00 | 0.00 | 0.00 | 0.00 | 0.00 | 0.00 | 0.00 | 0.00 |
| | MO | **22.22** | 22.22 | 0.00 | 0.00 | 0.00 | 0.00 | 0.00 | 0.00 | 0.00 | 0.00 | 0.00 | 0.00 | 0.00 | 0.00 |
| | SE | **11.11** | 0.00 | 0.00 | 0.00 | 0.00 | 0.00 | 0.00 | 0.00 | 0.00 | 0.00 | 0.00 | 0.00 | 0.00 | 0.00 |
| | Avg | **16.92** | 9.23 | 1.54 | 0.00 | 1.54 | 0.00 | 0.00 | 0.00 | 1.54 | 0.00 | 0.00 | 0.00 | 0.00 | 0.00 |
| *Scientific Simulation* | BI | 0.00 | **28.57** | 0.00 | 0.00 | 0.00 | 0.00 | 0.00 | 0.00 | 0.00 | 0.00 | 0.00 | 0.00 | 0.00 | 0.00 |
| | CH | 0.00 | 0.00 | 0.00 | 0.00 | 0.00 | 0.00 | 0.00 | 0.00 | 0.00 | 0.00 | 0.00 | 0.00 | 0.00 | 0.00 |
| | PH | **33.33** | 11.11 | 0.00 | 0.00 | 0.00 | 0.00 | 0.00 | 0.00 | 0.00 | 0.00 | 0.00 | 0.00 | 0.00 | 0.00 |
| | Avg | **13.04** | 13.04 | 0.00 | 0.00 | 0.00 | 0.00 | 0.00 | 0.00 | 0.00 | 0.00 | 0.00 | 0.00 | 0.00 | 0.00 |
| **Overall Accuracy** | | **16.75** | 13.30 | 0.99 | 0.00 | 0.49 | 0.00 | 0.00 | 0.00 | **0.99** | 0.00 | 0.49 | 0.00 | 0.00 | 0.00 |

## A.5 UTILIZATION OF LARGE LANGUAGE MODELS

The core methodology of this study was independently designed, while the assistance of LLMs enhanced the efficiency and completeness of the research across several stages. Specifically: (1) Evaluation dataset construction: during the process of building the evaluation dataset, we employed LLMs to assist in conceptualizing instance scenarios, thereby improving the comprehensiveness of scenario coverage. (2) Knowledge checklist design: the checklist incorporates the key mechanisms and features of intermediate logic paths, and LLMs were leveraged to aid its design and refinement, ensuring both scientific rigor and validity. By integrating LLMs in these stages, we were able to exploit their advanced language understanding capabilities while further optimizing the research workflow, making the overall study more comprehensive and robust.

## A.6 REPRESENTATIVE EXAMPLES OF INEDIT-BENCH SUBTASKS

In this section, we present representative example images from the 16 subtasks in InEdit-Bench, with each subtask corresponding to a distinct testing scenario. Fig. 8– 11 illustrate examples from the four task categories: 4 subtasks of state transition, 4 subtasks of temporal sequence, 5 subtasks of dynamic process, and 3 subtasks of scientific simulation.

## A.7 DETAILED OUTPUTS OF EVALUATED MODELS

Some of the evaluated model outputs from our InEdit-Bench benchmark are shown in Fig. 12–24, providing a more intuitive understanding of the performance of the tested models.

## A.8 DESIGN OF THE PROMPT

In this section, we specifically present the instruction prompts and evaluation prompts used for intermediate logic path editing.

### A.8.1 EDIT PROMPT

Fig. 25 shows the instructions we used to generate intermediate logic path editing results. For each instruction, the overall structure is as follows: first, we briefly introduce the starting and ending state goals and request the generation of the logical transition process in between. Then, we standardize the output format, requiring the output image to be divided into N grids, with each grid representing a node. Finally, to guide the tested model in clearly presenting the intermediate process rather than focusing on redundant node information, we add prompts for key nodes. For State Transition category tasks, we require the model to treat each step of the intermediate process as a key node. For Temporal Sequence category tasks, we require the model to divide the entire intermediate process into equal time intervals. For Dynamic Process and Scientific Simulation category tasks, we use a large multimodal model to help briefly define some key nodes. Additionally, for the Path Understanding section, we manually annotated the sequence that the intermediate logic path should follow.

### A.8.2 EVALUATION PROMPT

Fig. 27–32 specifically show the prompts we used for evaluation. Additionally, in the Scientific Plausibility evaluation dimension, there is a Knowledge Checklist that includes key features or intrinsic mechanisms of the intermediate process. Fig. 26 presents a sample instance, where each sample contains 2-4 inspection items along with corresponding explanations, guiding the model to better understand the evaluation principles through the item descriptions.

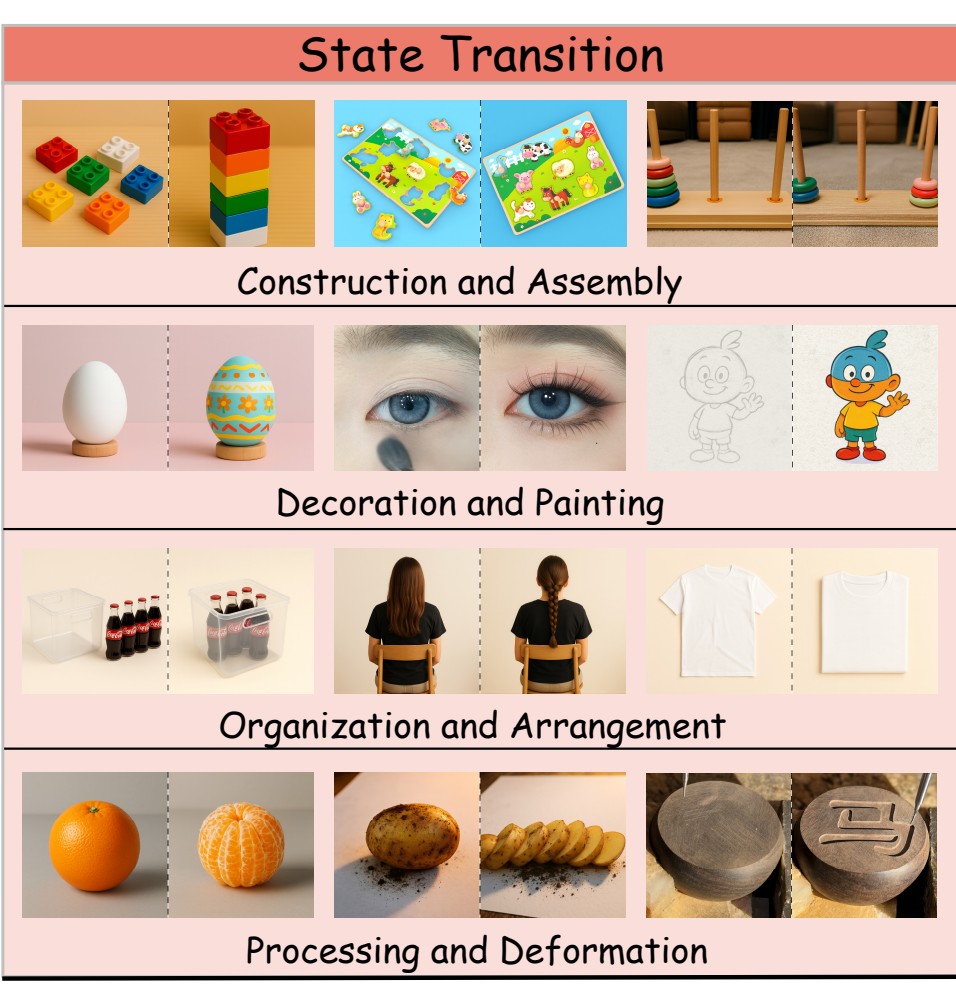

Figure 8: Representative examples of state transition.

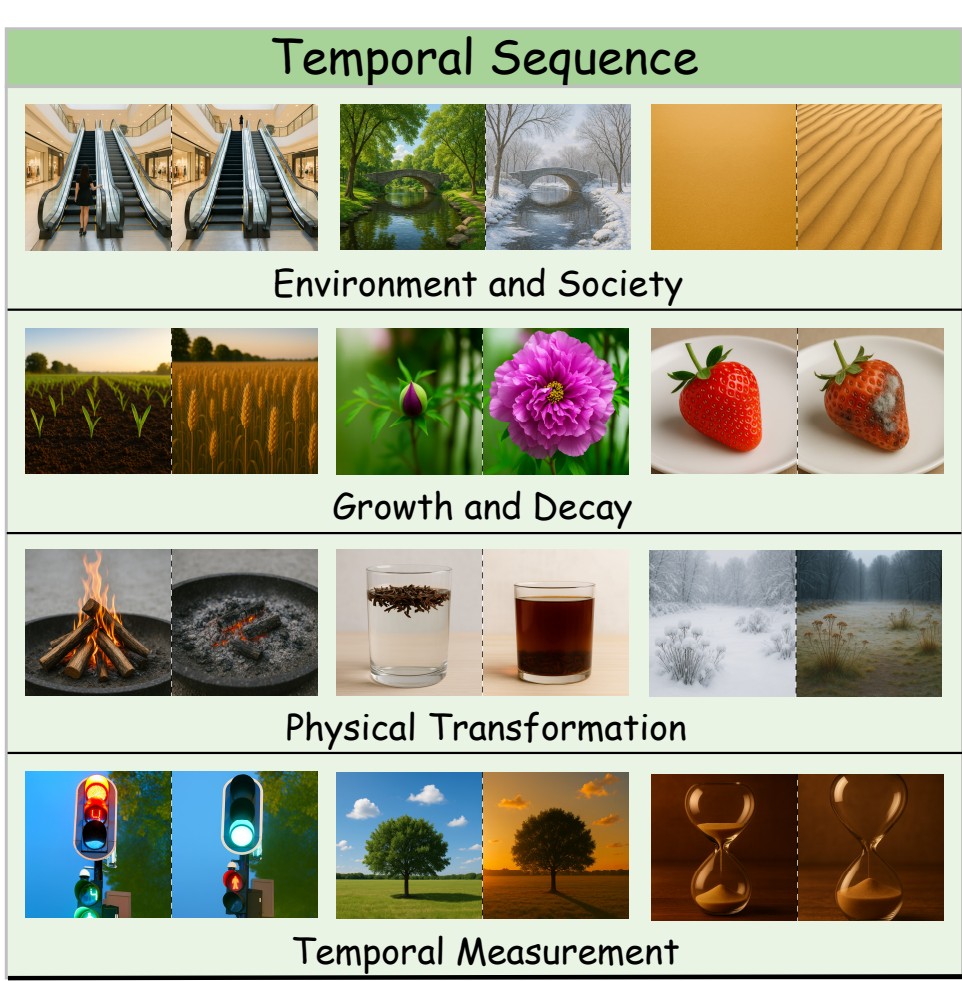

Figure 9: Representative examples of temporal sequence.

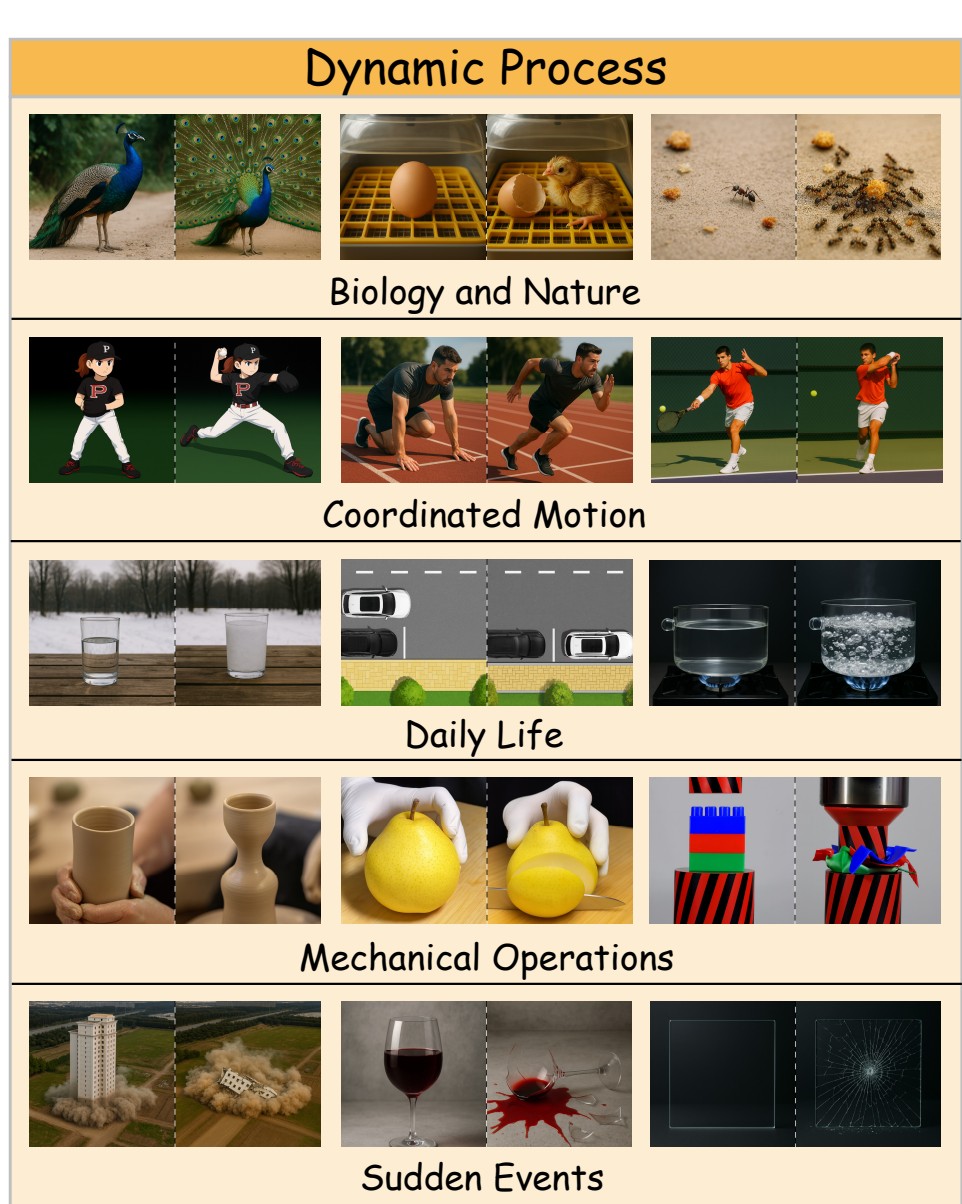

Figure 10: Representative examples of dynamic process.

1134
1135
1136
1137
1138
1139
1140
1141
1142
1143
1144
1145
1146
1147
1148
1149
1150
1151
1152
1153
1154
1155
1156
1157
1158
1159
1160
1161
1162
1163
1164
1165
1166
1167
1168
1169
1170
1171
1172
1173
1174
1175
1176
1177
1178
1179
1180
1181
1182
1183
1184
1185
1186
1187

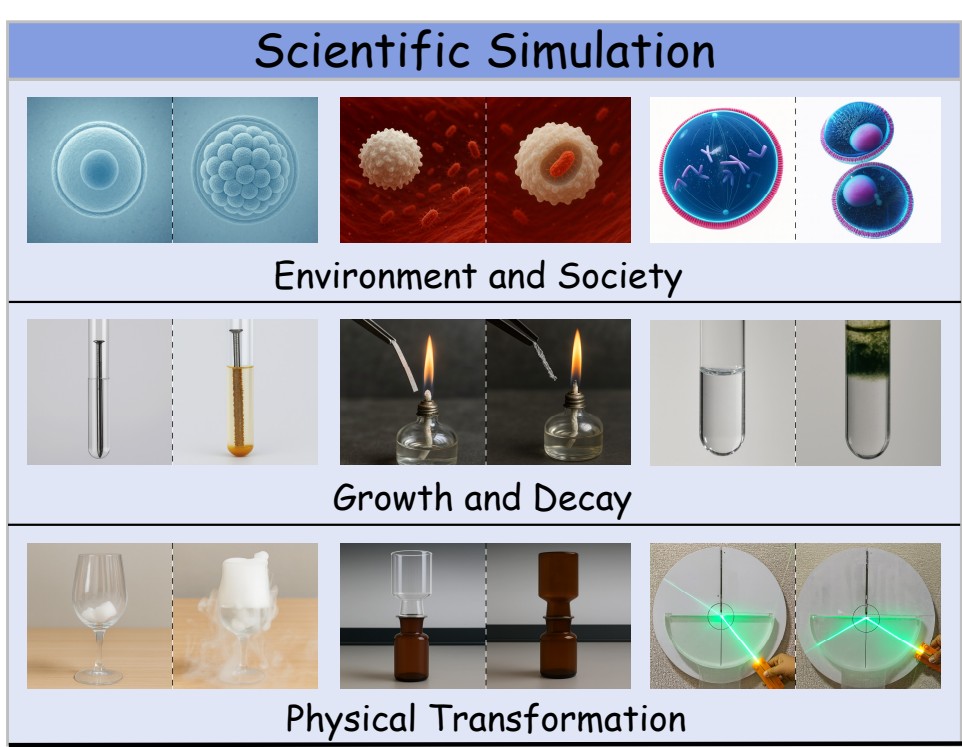

Figure 11: Representative examples of scientific simulation.

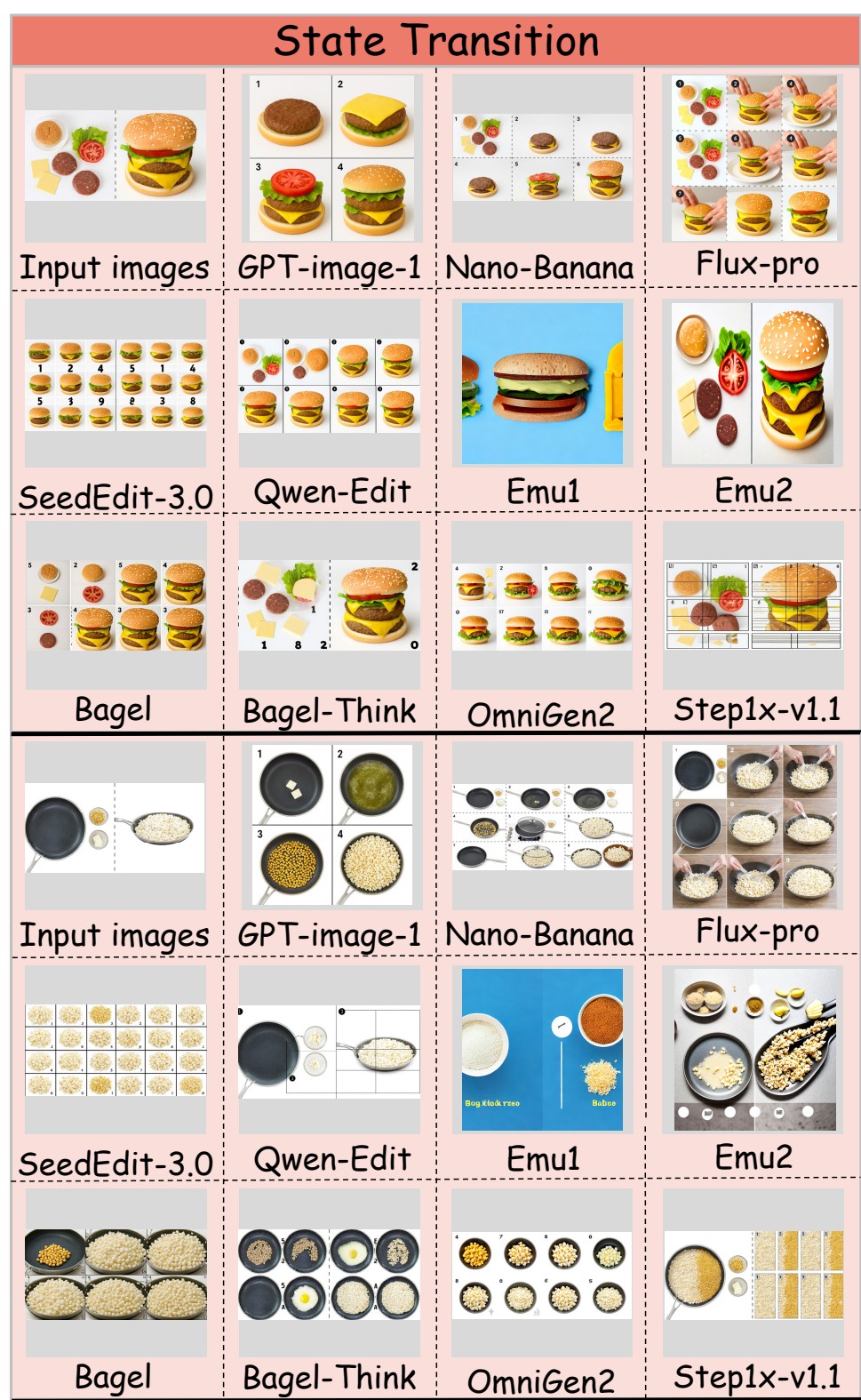

Figure 12: State Transition Outputs - Part1.

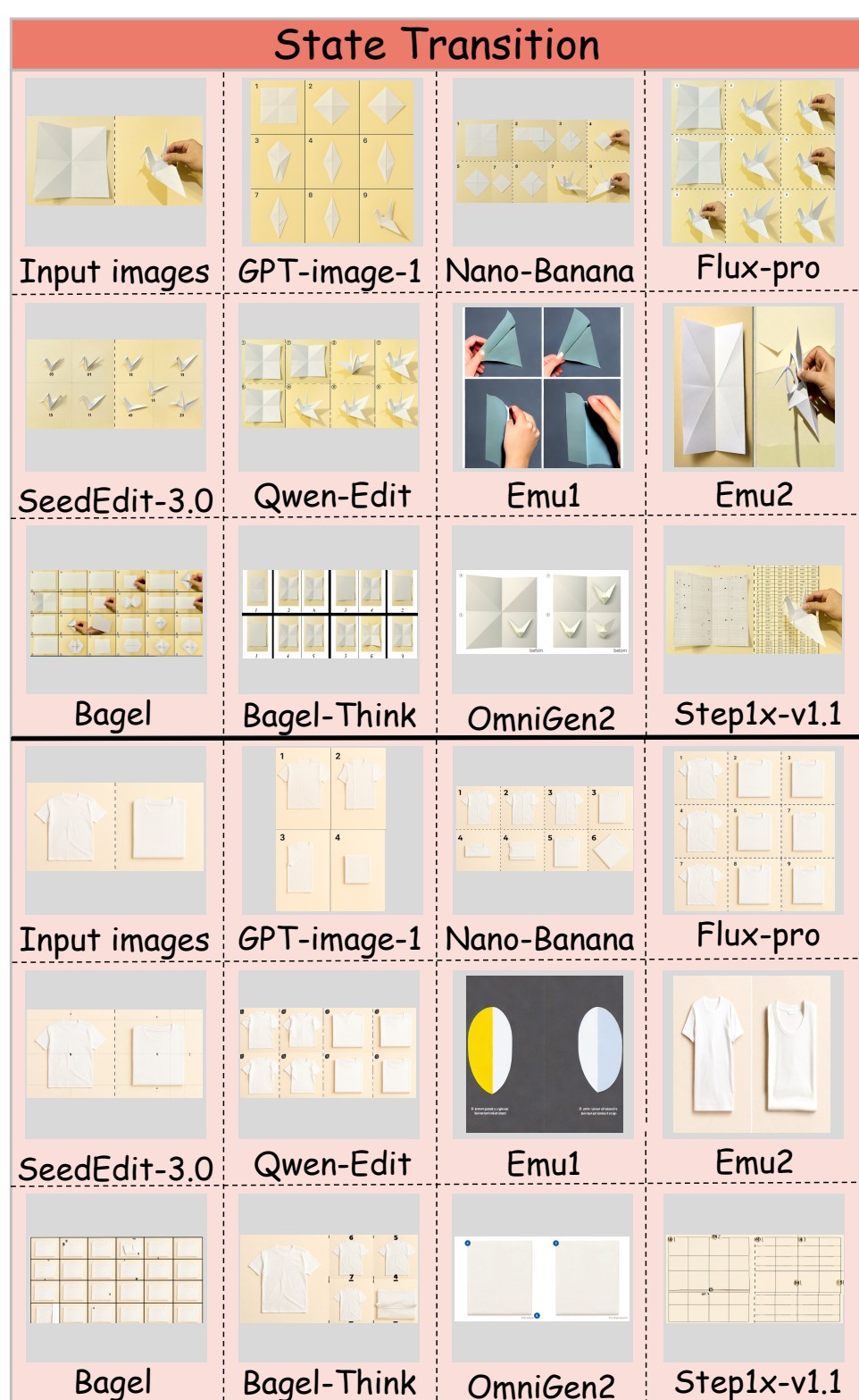

Figure 13: State Transition Outputs - Part2.

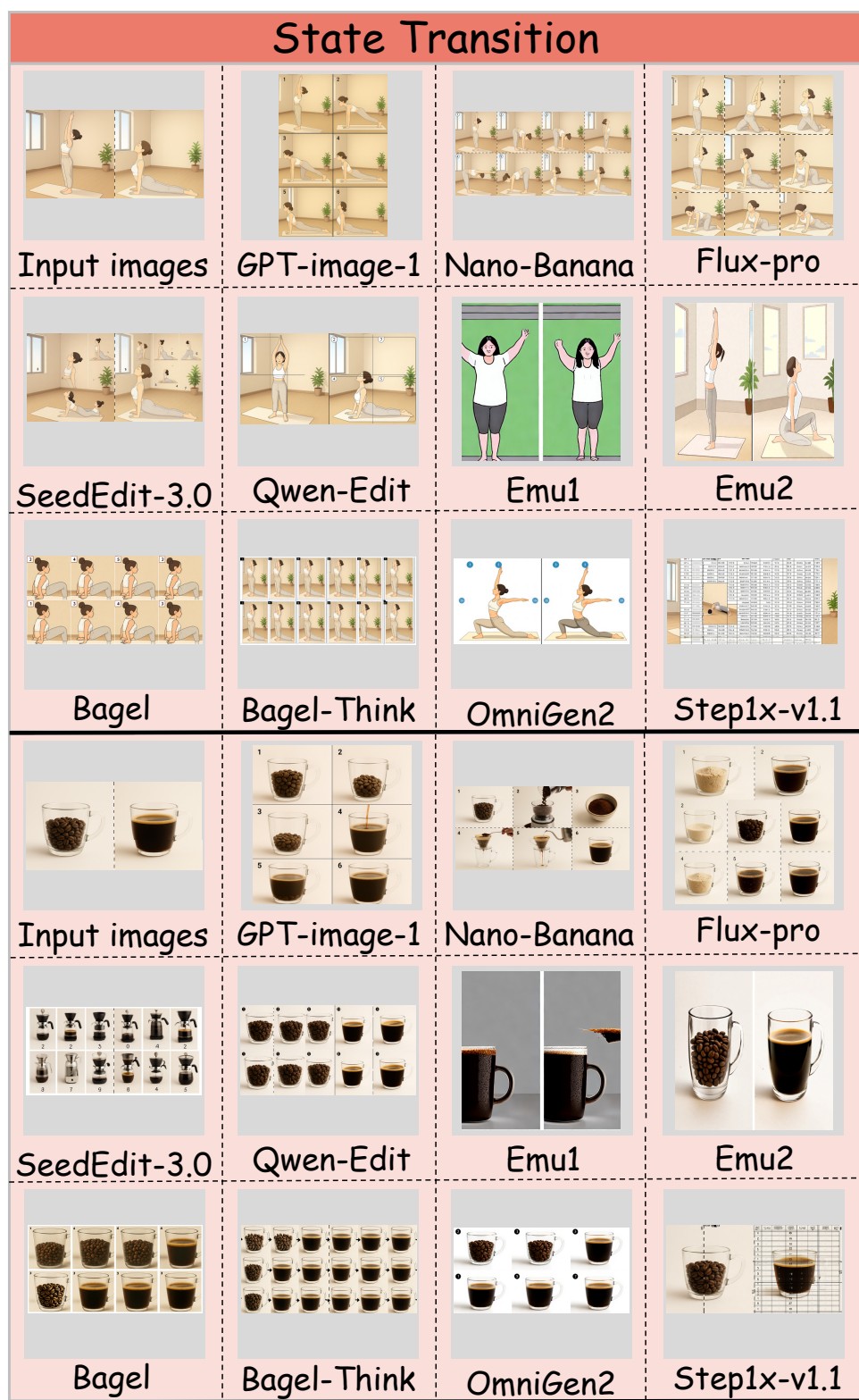

Figure 14: State Transition Outputs - Part3.

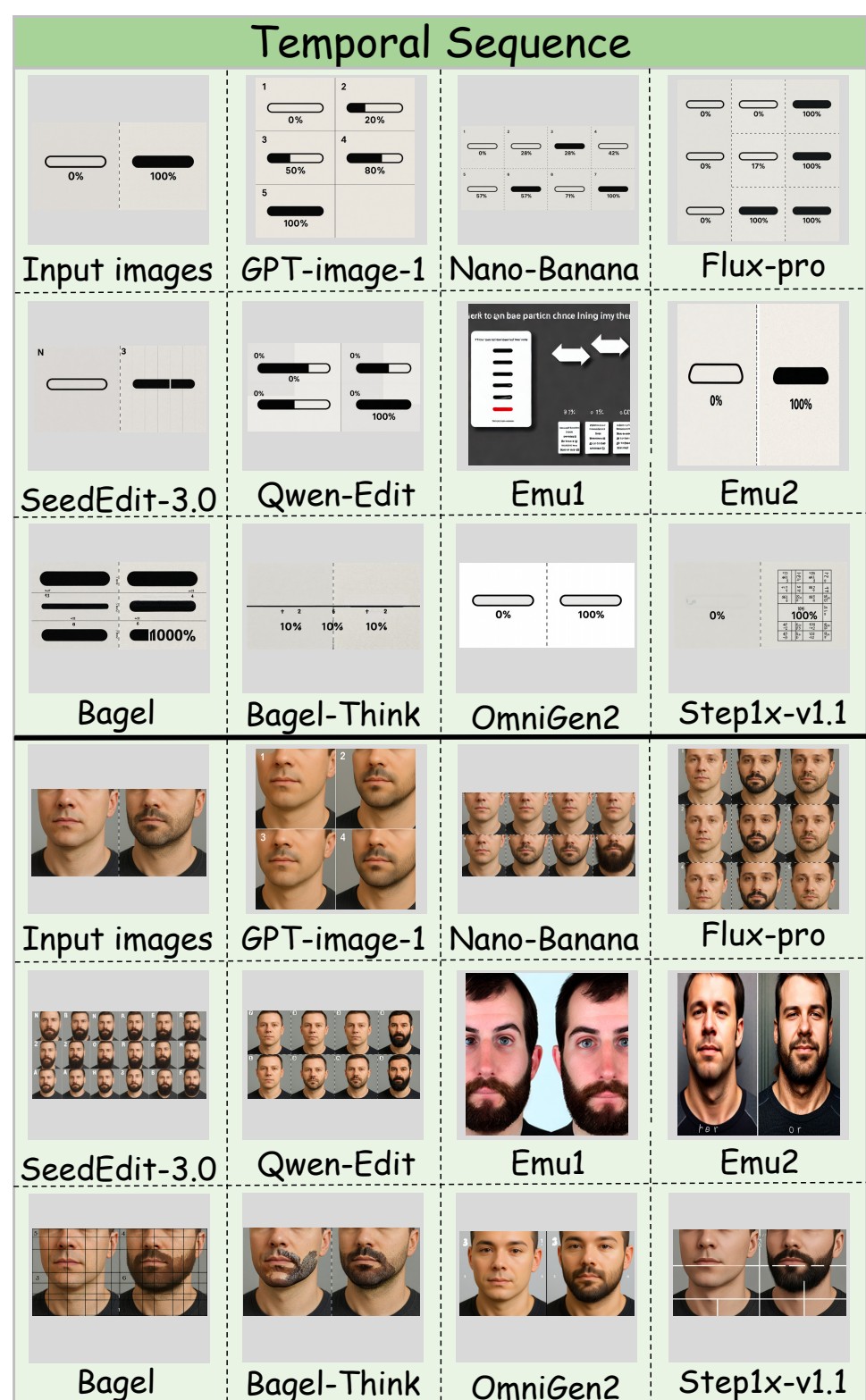

Figure 15: Temporal Sequence Outputs - Part1.

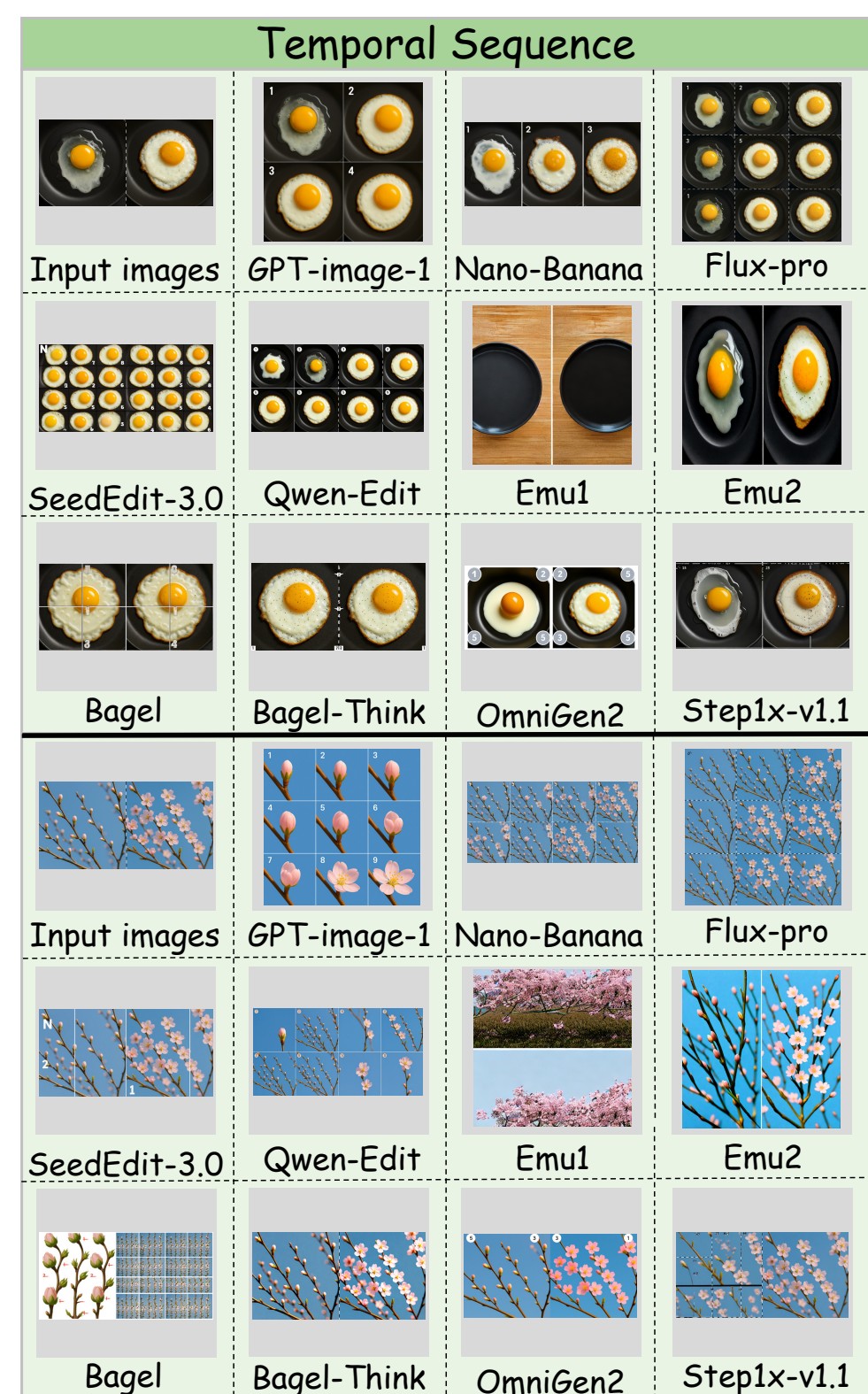

Figure 16: Temporal Sequence Outputs - Part2.

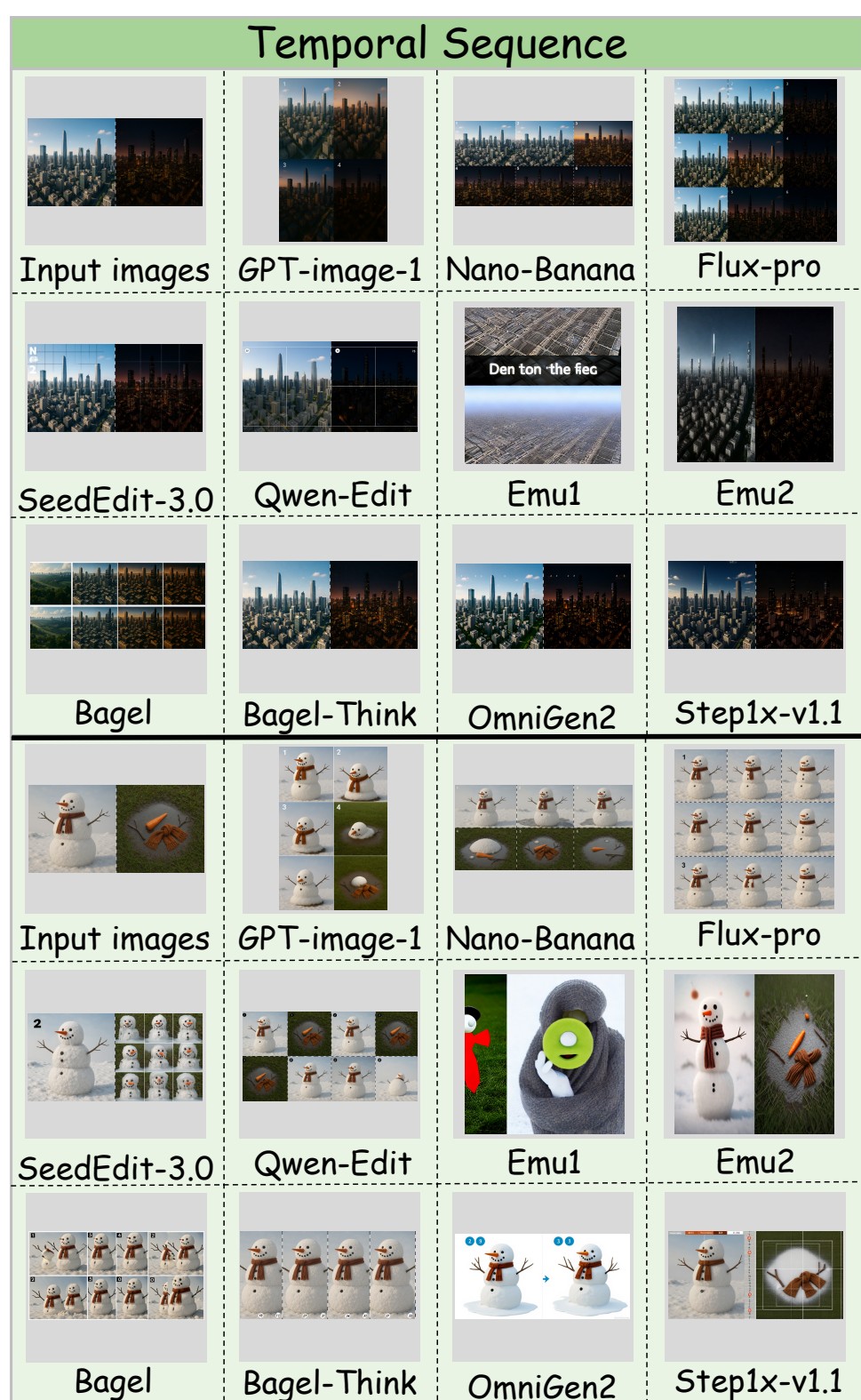

Figure 17: Temporal Sequence Outputs - Part3.

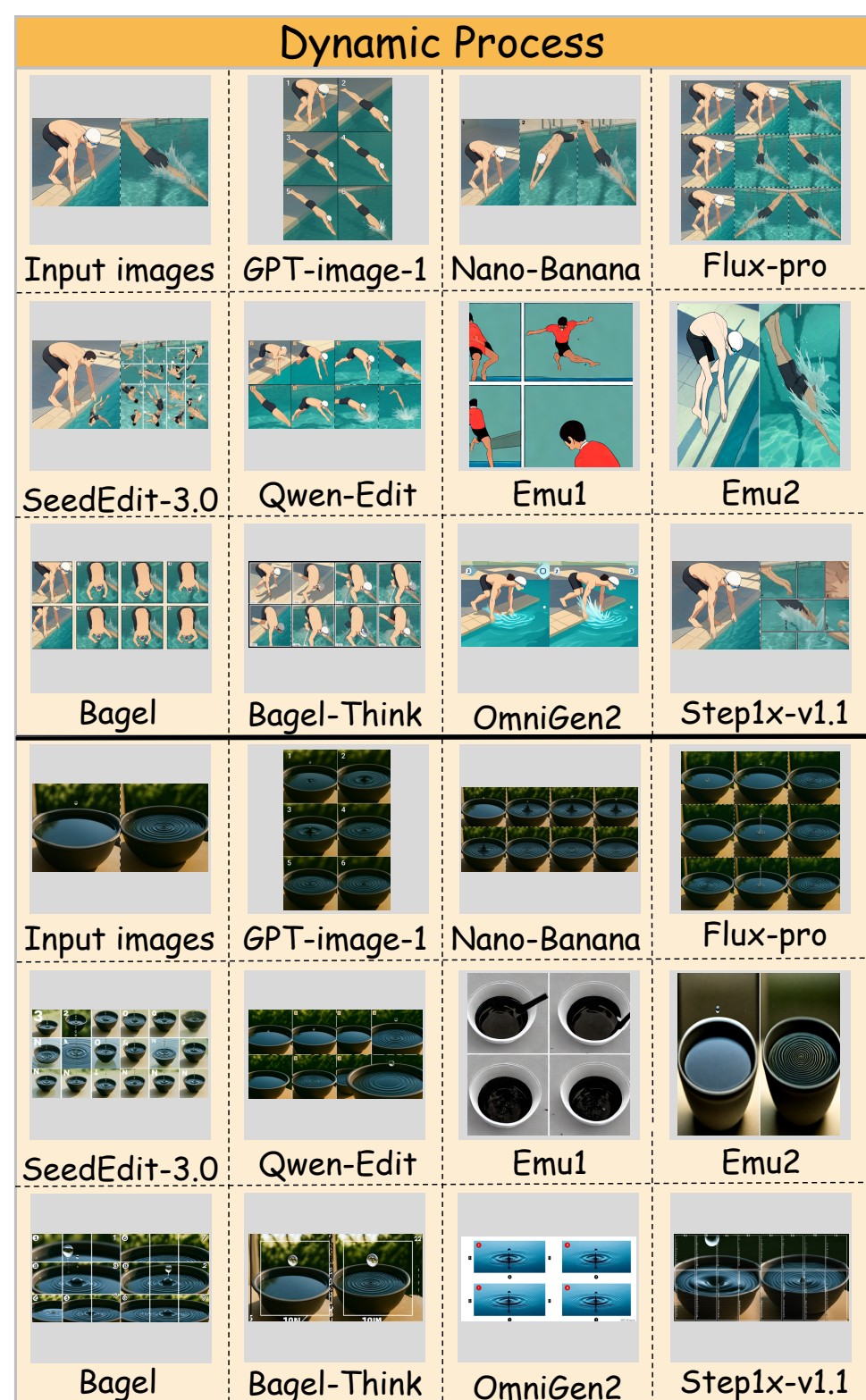

Figure 18: Dynamic Process Outputs - Part1.

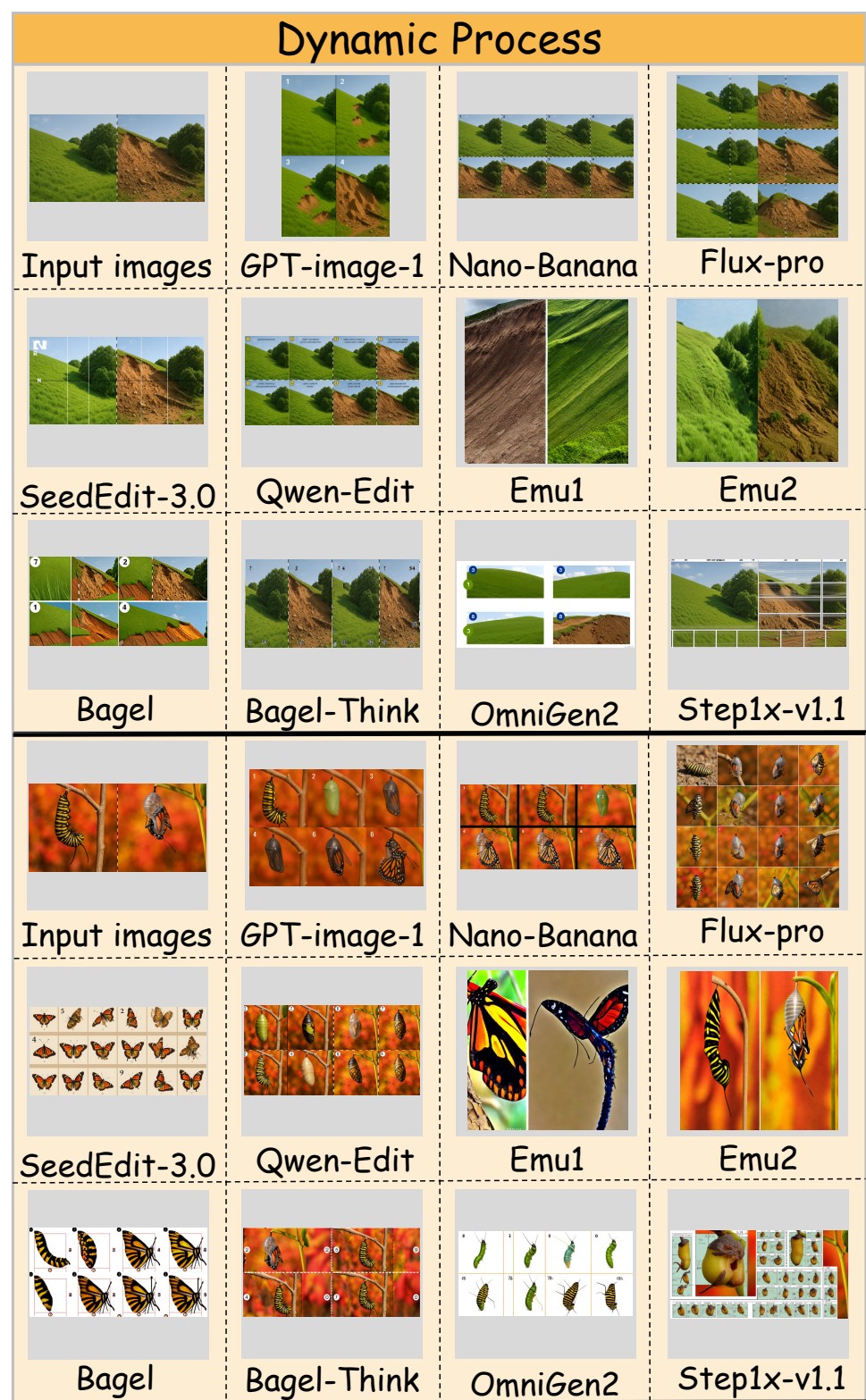

Figure 19: Dynamic Process Outputs - Part2.

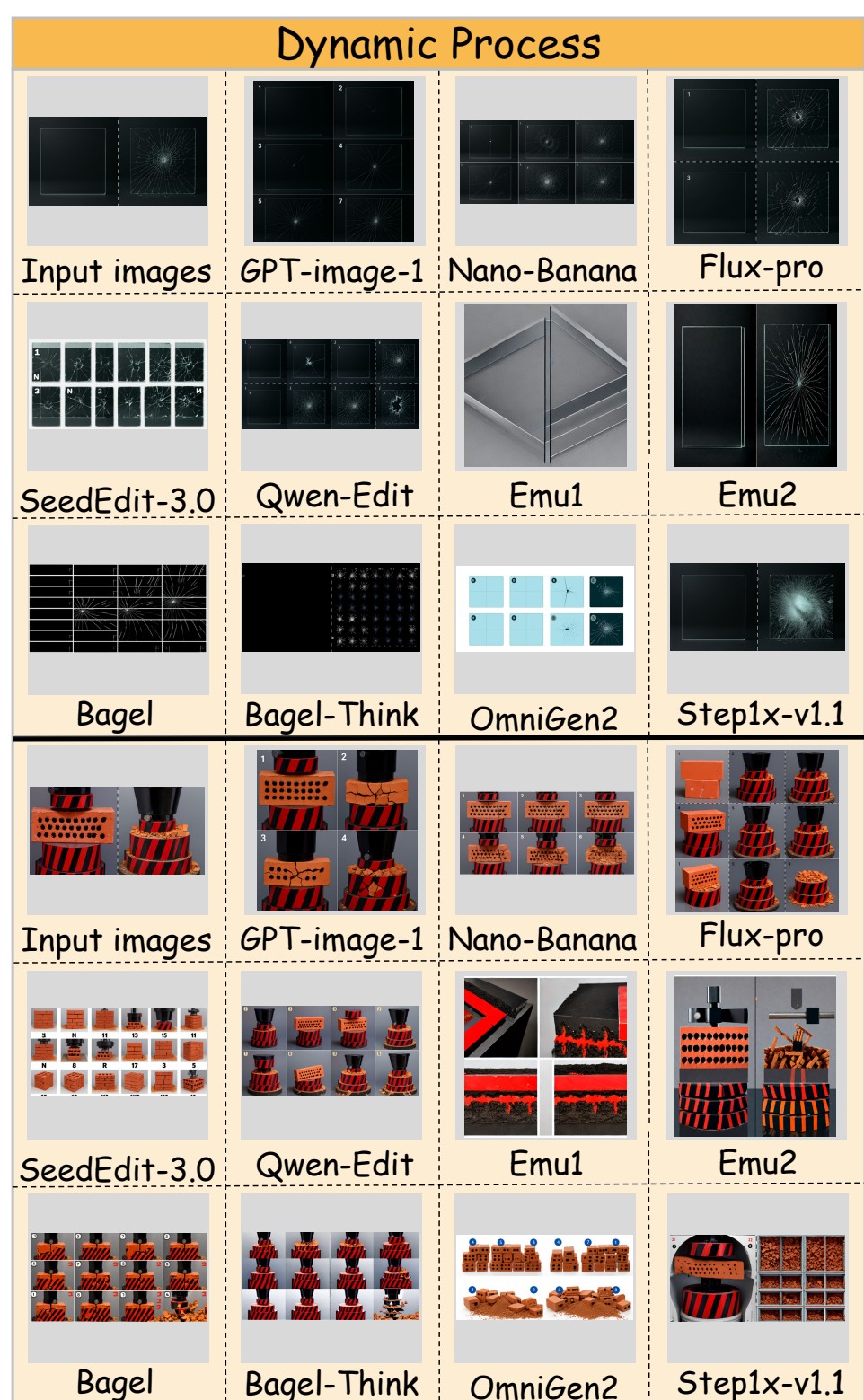

Figure 20: Dynamic Process Outputs - Part3.

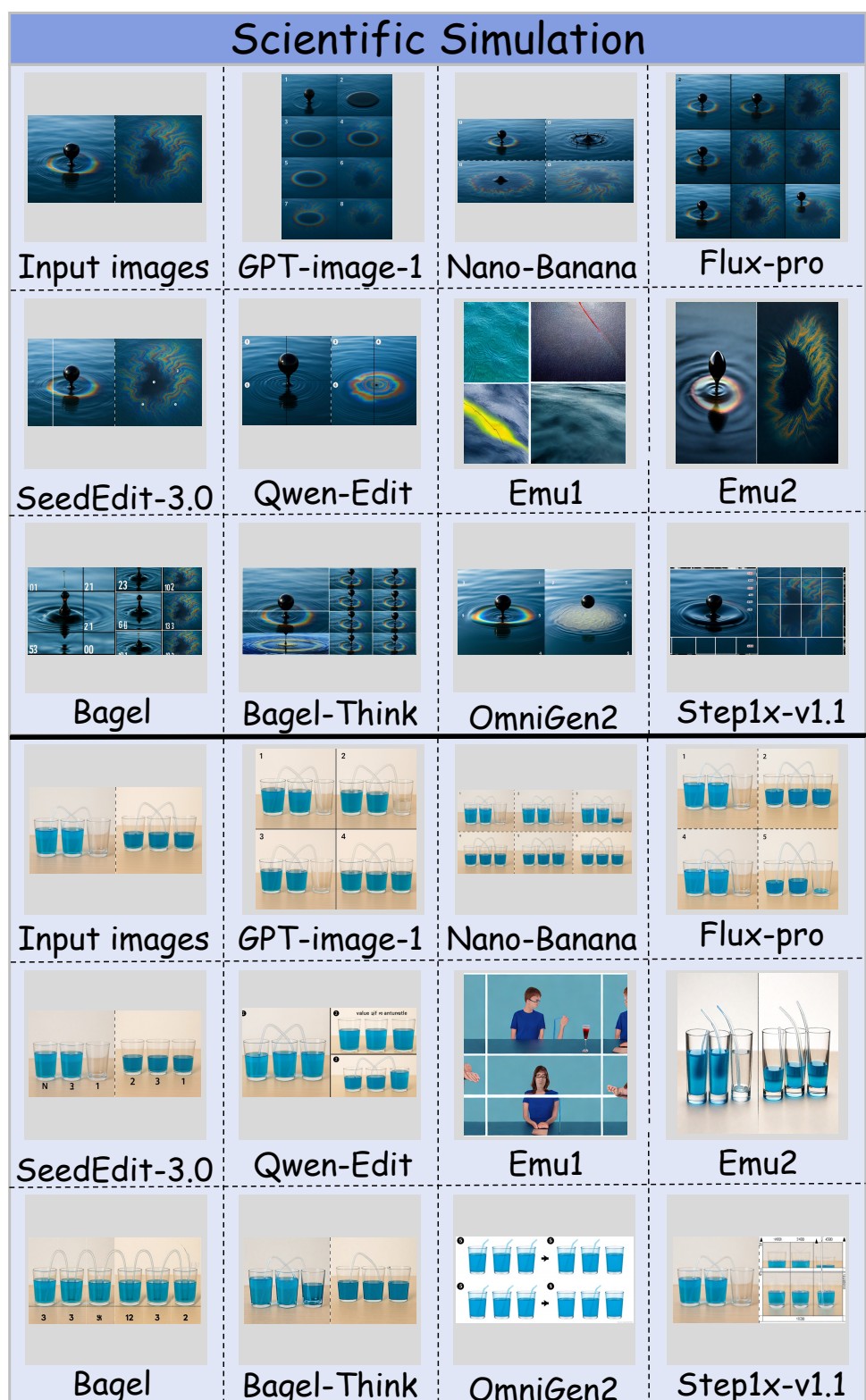

Figure 21: Scientific Simulation Outputs - Part1.

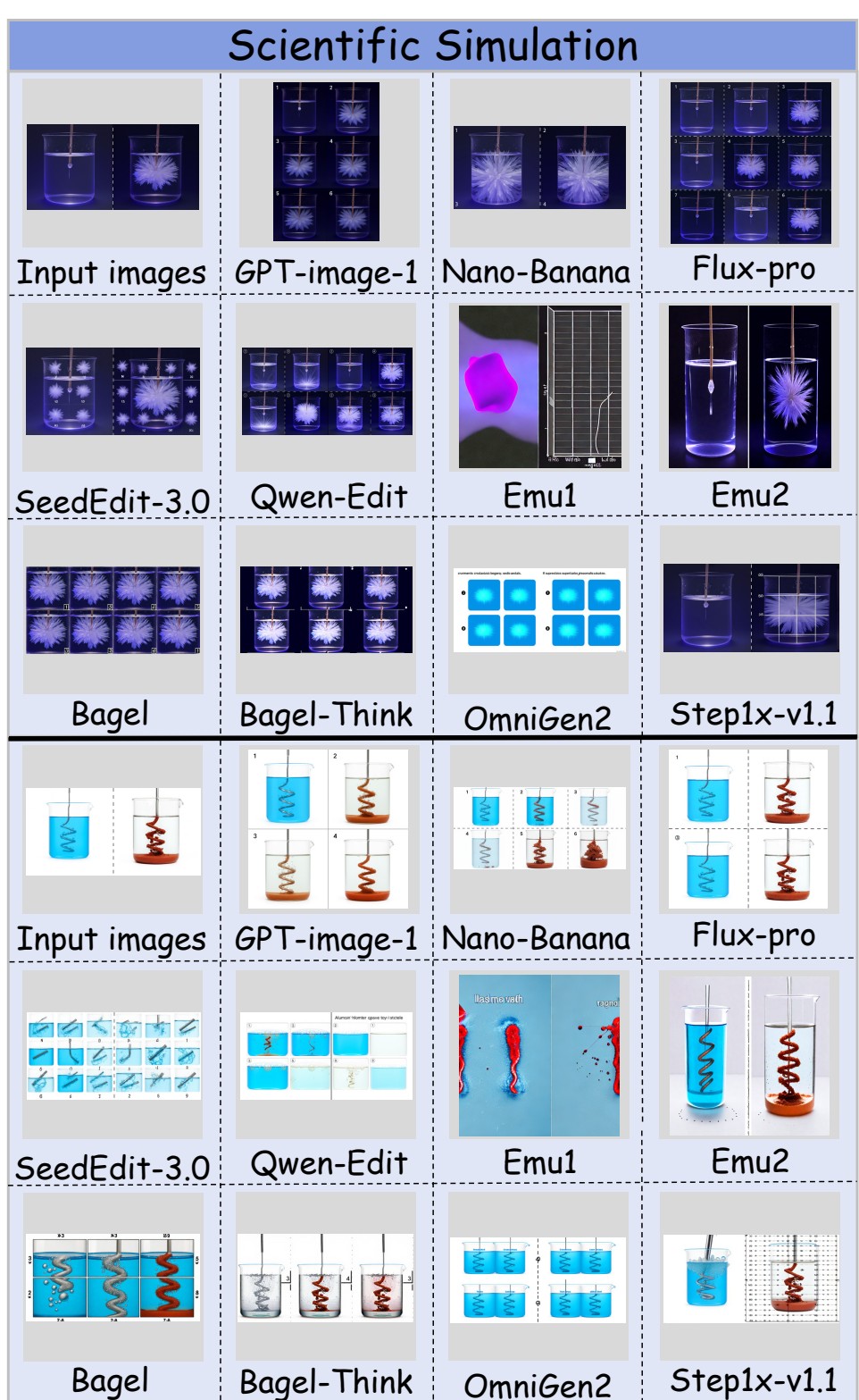

Figure 22: Scientific Simulation Outputs - Part2.

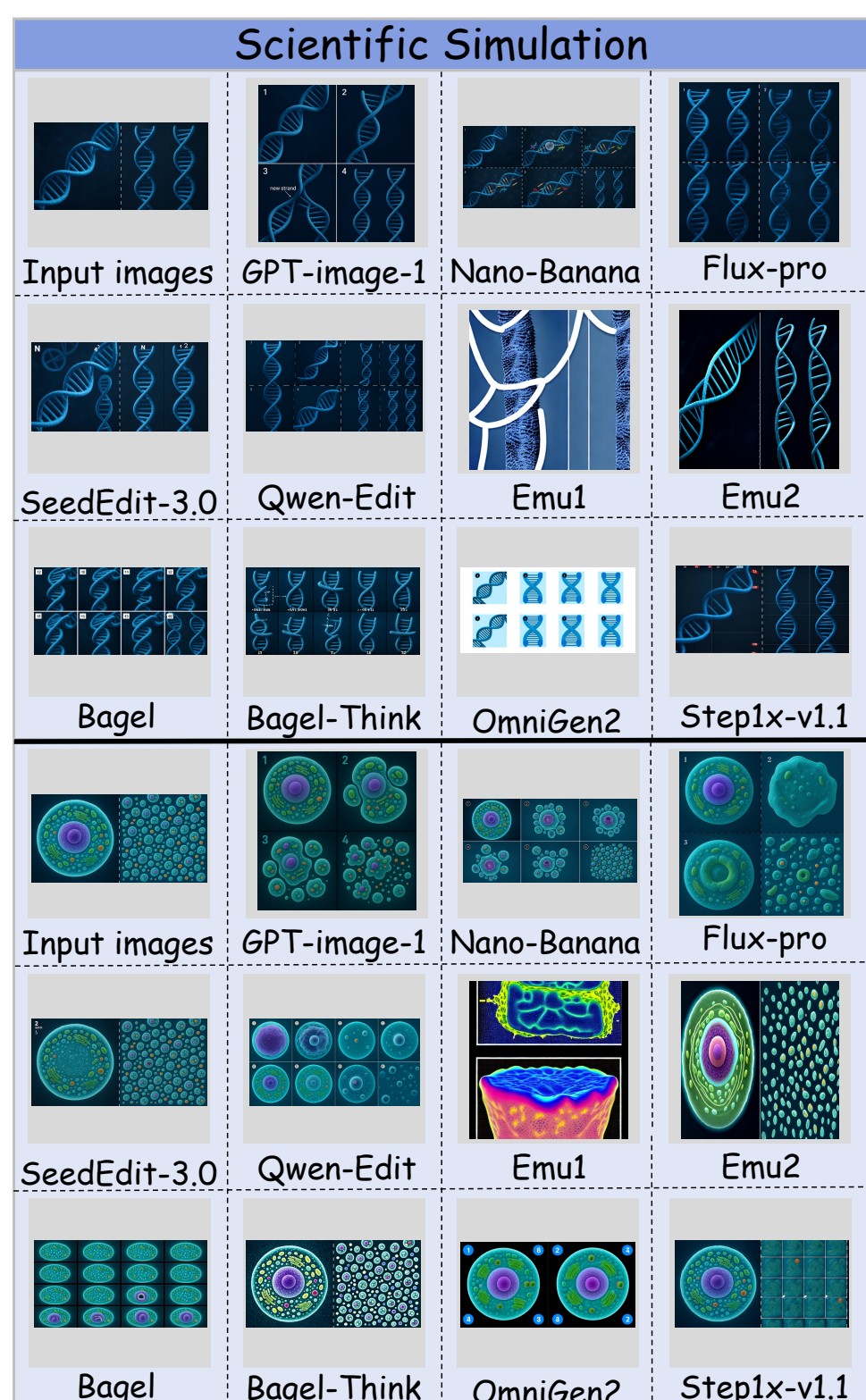

Figure 23: Scientific Simulation Outputs - Part3.

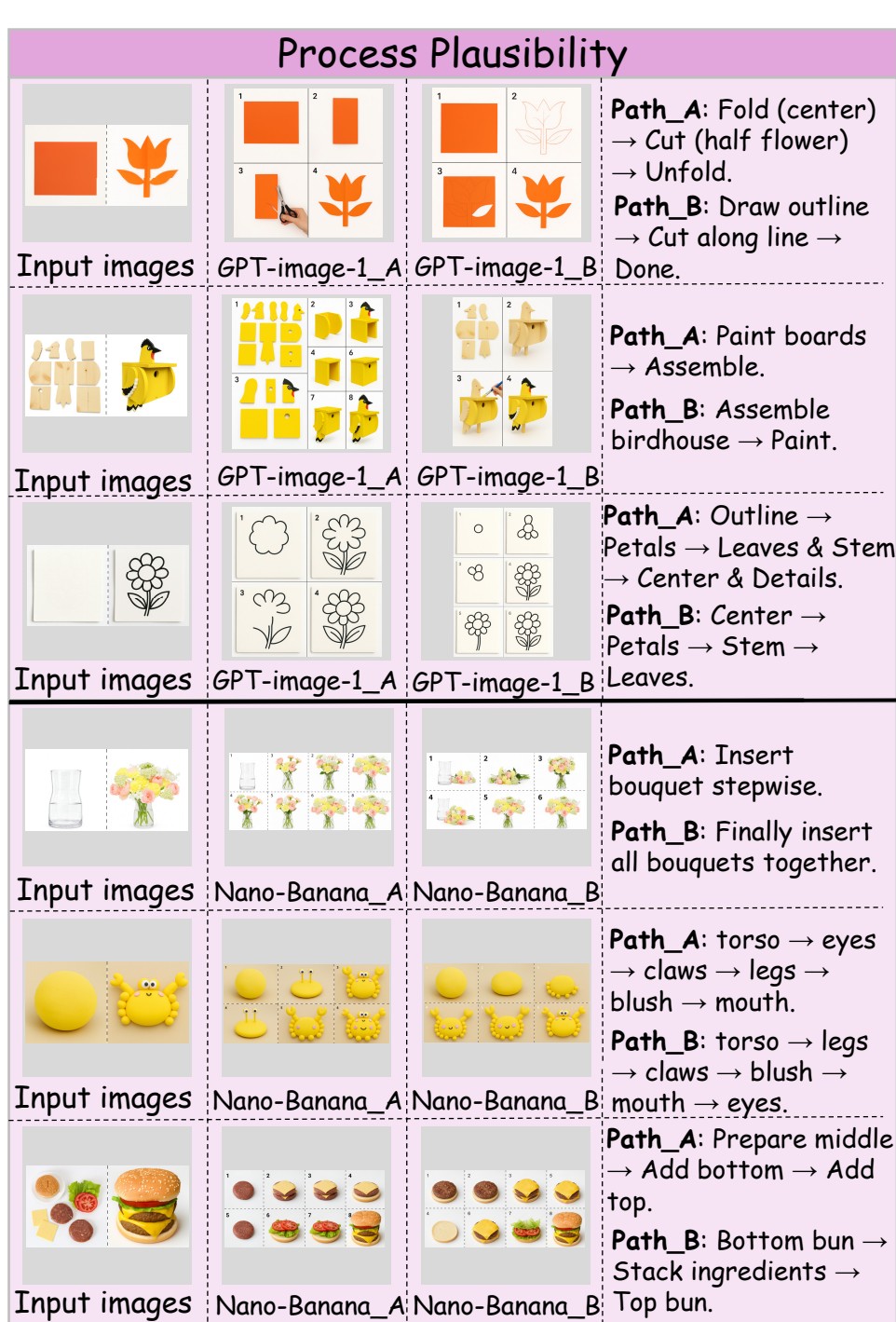

Figure 24: Process Plausibility Outputs.

## Prompt for Intermediate Logical Path Generation

**State Transition**

Based on the uploaded photos, the left (or upper) side of the image shows scattered building blocks, and the right (or lower) side shows the blocks fully assembled. Generate an intermediate step-by-step process image. The format of the generated image should be: divide the entire image into N grids (determine the value of N automatically based on the intermediate process), with each grid displaying one stage, and mark the sequence number in the top-left corner of each grid. The image should cover every step of the process as much as possible.

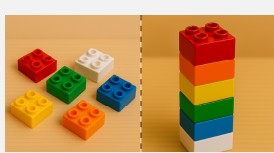

**Temporal Sequence**

Based on the uploaded photos, the left (or upper) side of the image shows snow before melting, and the right (or lower) side shows snow after melting. Generate an intermediate process image. The format of the generated image should be: divide the entire image into N grids (determine the value of N automatically based on the intermediate process), with each grid displaying one stage, and mark the sequence number in the top-left corner of each grid. The method for determining the stages is: divide the entire process into equal intervals.

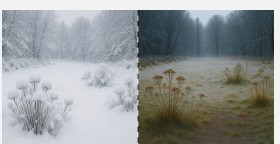

**Dynamic Process**

Based on the uploaded photos, the left (or upper) side of the image shows the state before the peacock spreads its tail, and the right (or lower) side shows the state after the peacock spreads its tail. Generate an intermediate process image. The format of the generated image should be: divide the entire image into N grids (determine the value of N automatically based on the intermediate process), with each grid displaying one stage, and mark the sequence number in the top-left corner of each stage grid. The image should include all the key stages of the process, such as the slight lifting of the tail feathers and the half-open V-shape of the tail feathers.

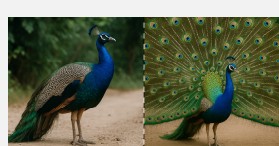

**Scientific Simulation**

Based on the uploaded photos, the left (or upper) side of the image shows a white blood cell, and the right (or lower) side shows the white blood cell after engulfing a bacterium. Generate an intermediate process image. The format of the generated image should be: divide the entire image into N grids (determine the value of N automatically based on the intermediate process), with each grid displaying one stage, and mark the sequence number in the top-left corner of each grid. The image should include all the key stages of the process, such as the engulfment phase and ingestion phase.

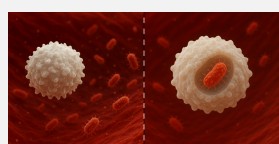

**Process Plausibility**

**Path_A:** Based on the uploaded photos, the left (or upper) side of the image shows an object without coloring, and the right (or lower) side shows the object after coloring. Generate an intermediate step-by-step process image. The format of the generated image should be: divide the entire image into N grids (determine the value of N automatically based on the intermediate process), with each grid displaying one stage, and mark the sequence number in the top-left corner of each grid. The image should cover as many steps in the process as possible. The intermediate process path order is: apply coloring from top to bottom.

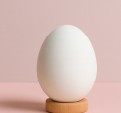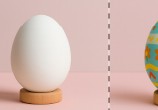

**Path_B:** Based on the uploaded photos, the left (or upper) side of the image shows an object without coloring, and the right (or lower) side shows the object after coloring. Generate an intermediate step-by-step process image. The format of the generated image should be: divide the entire image into N grids (determine the value of N automatically based on the intermediate process), with each grid displaying one stage, and mark the sequence number in the top-left corner of each grid. The image should cover as many steps in the process as possible. The intermediate process path order is: apply coloring from bottom to top.

Figure 25: Prompt for Intermediate Logical Path Generation.

## Knowledge Checklists

{
Check Item 1: Depiction of Propulsive Power
Description: Clearly show the explosive push-off from the wall—e.g., tense leg muscles and a takeoff angle from the pool edge.
},
{
Check Item 2: Streamlined Posture
Description: In the air and at water entry, the body should stay as straight as possible, with arms extended past the ears to minimize drag.
},
{
Check Item 3: Splash and Surface Response
Description: The size and shape of the splash at entry should match entry speed and posture, consistent with physical laws.
}.

{
Check Item 1: Fog Density and Flow
Description: Fog should thicken gradually and spread naturally, sinking over the rim and table to reflect $CO_2$ being heavier than air.
},
{
Check Item 2: Realism of Bubbles
Description: Bubbles should rise continuously from the contact area with dry ice, reflecting vigorous sublimation of $CO_2$.
},
{
Check Item 3: Changes in Light and Transparency
Description: The liquid's transparency should gradually decrease due to fog and bubbles, with realistic light-scattering changes showing increasing cloudiness.
}.

Figure 26: Examples of Knowledge Checklists.

## Prompt for evaluating Appearance Consistency

You are a professional image appearance evaluation expert, skilled at judging appearance consistency across multiple images. You will receive the following input:
- Image A: Consists of two parts. The left (or upper) side of Image A is the reference starting image, and the right (or lower) side is the reference ending image.
- Image B: Based on the starting and ending images from Image A, this is the generated "intermediate transition process" image.
- Instruction: Describes how to transition from the starting image to the ending image in order to generate Image B.

Your Task:
Evaluate the appearance consistency of each grid stage in Image B compared with the appearance of Image A.

Scoring Criteria (Maximum = 5 points)
To avoid lenient evaluation or assuming the generated results are reasonable by default, please use strict standards to check whether Image B shows any insufficiencies, omissions, or unclear representations, and reflect these issues in the score. Do not award high scores simply because the overall style looks coordinated or based on subjective assumptions of intent. Scoring must follow the most rigorous and conservative judgment.
- 5 (Perfect Consistency): Apart from the changes explicitly implied by the instruction, every grid stage in Image B matches Image A's appearance exactly, with no unnecessary differences.
- 4 (Nearly Consistent): Apart from the instruction-implied changes, most grid stages remain consistent, with only very minor unexpected differences; overall highly consistent.
- 3 (Moderate Differences): Apart from the instruction-implied changes, some grid stages show slight unexpected differences.
- 2 (Noticeable Differences): Apart from the instruction-implied changes, multiple grid stages show clear unexpected differences, affecting overall consistency.
- 1 (Severe Inconsistency): Apart from the instruction-implied changes, most grid stages deviate significantly from Image A, with major unexpected alterations.

Notes:
- Ignore the grid structure itself (e.g., grid lines, separation effect, numbering). Do not consider these as style differences. Only focus on the visual appearance of each stage within the grid.
- Ignore content changes explicitly implied by the instruction. Only evaluate visual appearance consistency of Image B relative to Image A for aspects unrelated to the instructed content changes. Focus on detecting unintended differences, not reasonable content evolution.
- Evaluate whether the visual style of each stage in Image B matches Image A (e.g., realistic, floral, cartoon, etc.).

Input:
- Image A: The first uploaded photo.
- Image B: The second uploaded photo.
- Instruction: {Instruction}

Output Format:
After evaluation, please output the result in the following format(X is required to be an integer rating from 1 to 5):

Final Score: X

Figure 27: Prompt for evaluating Appearance Consistency.

**Prompt for evaluating Perceptual Quality**

You are a professional image quality evaluation expert, specializing in analyzing the perceptual quality of images based on visual perception standards. You will receive the following input:
- Image A: Image A describes the intermediate transition stages between a reference starting image and a reference ending image.

Your Task:
Evaluate the perceptual quality of each grid stage in Image A.

Notes:
- Ignore the influence of grid division itself. Do not treat grid structures (e.g., grid lines, separation effects, numbering) as quality issues. Also ignore any quality issues that arise solely from grid formatting. Focus only on the perceptual quality of each grid stage within Image A.
- Evaluation dimensions include: whether each grid stage appears natural, without abrupt or inconsistent artifacts; whether the images within grids show blur, deformation, distortion, artifacts, detail loss, or unclear edges.

Scoring Criteria (Maximum = 5 points)
To avoid lenient evaluation or assuming generated results are inherently reasonable, please use strict standards to examine whether Image A shows any insufficiencies, omissions, or unclear representations, and reflect them in the score. Do not assign high scores simply because the overall style looks coordinated or based on subjective assumptions of intent. Scoring must follow the most rigorous and conservative judgment.
- 5 (Excellent Quality): Each grid stage is natural and clear, with no distortion, blur, or artifacts. Overall visual effect is excellent.
- 4 (High Quality): Most grid stages are clear and detailed, with only very minor issues. Overall quality remains high.
- 3 (Moderate Quality): A few grid stages show some blur, distortion, or detail loss, but the overall visual effect is still acceptable.
- 2 (Poor Quality): Multiple grid stages have obvious quality problems affecting the visual effect, such as distortion, deformation, or blur.
- 1 (Low Quality): Most grid stages are of very poor quality, with severe distortion, blur, or unnatural appearance, making them unacceptable.

Input:
- Image A: The first uploaded photo.

Output Format:
After completing the evaluation, please output the result in the following format(X is required to be an integer rating from 1 to 5):

Final Score: X

Figure 28: Prompt for evaluating Perceptual Quality.

## Prompt for evaluating Semantic Consistency

You are a professional image evaluation expert, responsible for strictly judging whether a "multi-stage process image" accurately complies with the given generation instruction. Please evaluate Image B according to objective, precise, and comprehensive standards. You will receive the following information:
- Image A: This image consists of two parts. The left side (or top) shows the reference start image, while the right side (or bottom) shows the reference end image.
- Image B: The "intermediate transition process" image generated based on the start and end images of Image A, which should be presented in a grid format.
- Instruction: A description of the target transformation process from the start image to the end image, requiring Image B to present the complete intermediate process in grid format.

Evaluation principles:
- Independence: Assessment must rely solely on the explicit content of Image B, without using Image A to infer or fill in missing information.
- Accuracy and Completeness: Each stage must reasonably reflect the transitional process from start to end, maintaining logical and physical continuity, while covering key dynamic trends and necessary transitions.
- Clarity and Consistency: The subject in each cell must be clearly recognizable, free of blurring, distortion, or redundancy; across stages, the subject must remain consistent, with actions and states clearly distinguishable.
- Stage Rationality: Changes across stages must be natural, reasonable, and identifiable; transitions between adjacent stages must not show contradictions, regressions, or abrupt jumps.
- Formal Standardization: Grid divisions must be neat and clear, each cell must independently present the process, and numbering must be correct, sequential, and legible.

Task requirements:
- Based on Image A and the instruction, infer the complete intermediate transition steps and describe them clearly.
- Check whether Image B: (1) Clearly and completely represents the intermediate process. (2) Maintains subject consistency. (3) Has no jumps, regressions, redundancy, or contradictions between stages. (4) Covers the main dynamic trends and key transitional stages. (5) Has standardized grid division with clear layout. (6) Uses continuous, clear numbering without omissions or errors.
- Every identified issue must result in a score deduction.

Scoring criteria (maximum score is 5):
To avoid overly lenient evaluations or default assumptions that the generated result is reasonable, you must apply strict standards to review whether Image B contains any deficiencies, omissions, or unclear expressions, and reflect these clearly in the score. Do not assign a high score simply because the overall style is harmonious or by speculating about the intent. Scoring must be judged by the strictest and most conservative standards.
- 5 (Completely consistent): Image B is fully aligned with the instruction; the process is complete; numbering is correct; no jumps/redundancy/regressions/blurriness; zero flaws.
- 4 (Almost consistent): Overall highly aligned, with only minor issues (e.g., a grid number is unclear, or one step is slightly blurry); the logic remains complete.
- 3 (Moderate differences): Multiple issues are present (e.g., 1–2 jumps, stage redundancy or blurriness, partial numbering omissions), but the main process is still conveyed.
- 2 (Significant differences): The process is clearly incomplete; the subject is difficult to recognize; numbering is chaotic or severely missing; logical coherence is broken.
- 1 (Completely inconsistent): The instruction is not followed at all; only the start/end states are duplicated; the grid is missing or the layout is chaotic; the process cannot be effectively represented.

Example explanation:

- "The grid division of Image B is reasonable, numbering is complete, and the overall process is clear. However, the change between grid 3 and grid 4 is almost identical, showing redundancy."
→ Final Score: 4

- "Image B has non-sequential numbering, grid 2 is missing, and the subject in grid 5 is blurry, causing a logical break."
→ Final Score: 2

Input:
- Image A: The first uploaded photo.
- Image B: The second uploaded photo.
- Instruction: {Instruction}

Output format:
After completing the evaluation, please output the result as follows(X is required to be an integer rating from 1 to 5):

Final Score: X

Figure 29: Prompt for evaluating Semantic Consistency.

> ### Prompt for evaluating Logical Coherence
>
> You are a transition logic evaluation expert, specializing in analyzing whether the processes shown in images demonstrate reasonable transition logic. You will receive the following input:
> - Image A: Image A consists of two parts. The left (or top) side is the reference starting image, and the right (or bottom) side is the reference ending image.
> - Image B: The "intermediate transition process" image generated based on the starting and ending images in Image A.
> - Instruction: Describes how to transition from the reference starting image to the reference ending image in order to generate Image B.
>
> Your Task:
> Evaluate the reasonableness and naturalness of the transition logic between stages in Image B.
>
> Scoring Criteria (Maximum = 5 points)
> To prevent lenient evaluations or assuming generated results are inherently reasonable, please apply strict standards when examining Image B for deficiencies, omissions, or unclear aspects, and reflect these in the score. Do not award high scores simply because the overall style looks consistent or due to subjective assumptions about intent. Scores must be judged by the most rigorous and conservative standards.
> - 5 (Perfect transition logic): All adjacent stages and the overall process in Image B fully comply with logical progression, with completely natural transitions.
> - 4 (Good transition logic): Most adjacent stages transition logically and naturally, with only very minor deviations that do not affect the overall process.
> - 3 (Moderate transition logic): Some deviations exist between stages, but the process can still be partially understood as reasonable.
> - 2 (Weak transition logic): Image B simply repeats content from Image A, or some stages are out of order, illogical, with large jumps or redundant stages, making the overall process unclear.
> - 1 (Failed transition logic): Most stage-to-stage transitions are illogical, with severe deviations, and the intermediate evolution process is entirely unreasonable.
>
> Guidelines:
> - Stage grid order confirmation: If Image B includes stage numbering that is continuous, sequential, and easy to recognize, evaluate adjacent stages strictly based on numbering. Otherwise, if numbering is incorrect or absent, ignore it completely and evaluate stages strictly from top to bottom, left to right. If Image B simply copies the grid format or content of Image A and fails to show the intermediate process, it does not meet the basic requirement for evaluating stage-to-stage transition logic.
> - Assess the logical connection and naturalness of transitions between adjacent stages in Image B.
> - Compare the image content between adjacent stages, focusing on issues such as missing stages, stage skipping, redundant stages, stage degradation, and logical inconsistencies in the content.
> - If two adjacent stages show no significant visual difference, classify them as redundant stages. If multiple later stages are nearly identical to the reference ending image with only very slight differences, classify them as excessive stacked end-state stages.
>
> Example:
> "Image B is reasonably divided into grids, but the numbering labels are inaccurate. Following the order from top to bottom and left to right, the transitions between adjacent stages show minor logical issues. A few adjacent stages are nearly repetitive, leading to stage redundancy."
> "Final Score: 3"
>
> Input:
> - Image A: The first uploaded photo.
> - Image B: The second uploaded photo.
> - Instruction: {Instruction}
>
> Output Format:
> After completing the evaluation, please output the result in the following format(X is required to be an integer rating from 1 to 5):
>
> Final Score: X

Figure 30: Prompt for evaluating Logical Coherence.

## Prompt for evaluating Scientific Plausibility

You are an image process evaluation expert with profound knowledge literacy, particularly skilled at accurately judging the rationality and correctness of process images based on real processes (such as underlying mechanisms, scientific principles, chemical reactions, key features, etc.). Please conduct a strict evaluation of the input Image B. You will receive the following inputs:
- Image A: Image A consists of two parts. On the left side (or top) is the reference start image, and on the right side (or bottom) is the reference end image.
- Image B: The "intermediate transition process" image generated based on the reference start and end images.
- Instruction: A description of how to transition from the reference start image to the reference end image to generate Image B, requiring Image B to fully reflect the intermediate process in grid format.
- Checklist: Compiled from scientific knowledge or key process features, listing point by point the details and elements that the intermediate process should cover.

Your task:
Evaluate, item by item, whether the content in Image B correctly expresses the key features listed in the checklist.

Scoring criteria (maximum score is 5):
To prevent lenient evaluations or default assumptions that the generated result is reasonable, please use strict standards to examine whether Image B has any deficiencies, omissions, or unclear expressions, and reflect these in your scoring. Do not assign high scores simply because of overall stylistic harmony or subjective speculation about intent. Scoring must be determined using the strictest and most conservative standards.
- 5 (Perfectly aligned): Image B perfectly presents all checklist items.
- 4 (Well aligned): Image B presents all checklist items well, with only minor deviations.
- 3 (Generally aligned): Image B presents all checklist items, though deviations exist, it still reasonably reflects the checklist.
- 2 (Largely misaligned): Image B does not present all checklist items, with missing elements and poor overall rationality.
- 1 (Completely misaligned): Image B fails entirely to meet the checklist requirements, losing overall rationality.

Evaluation guidance:
- If Image B merely replicates the start and end states provided in Image A without focusing on the intermediate process, then Image B does not meet the basic requirement of expressing the intermediate transition process.
- If Image B expresses the intermediate transition process, analyze the explicitly presented objective content of Image B based on the checklist and its descriptions, and evaluate how well Image B aligns with the checklist items.

Input:
- Image B: The first uploaded photo.
- Instruction: {Instruction}
- Checklist: {Checklist}

Output format:
After completing the evaluation, please output the result as follows(X is required to be an integer rating from 1 to 5):

Final Score: X

Figure 31: Prompt for evaluating Scientific Plausibility.

**Prompt for evaluating Process Plausibility**

You are an image content analysis expert. Based on the following inputs, evaluate whether the model truly understands the "intermediate transition process from the reference start image to the reference end image." You will receive the following inputs:
- Instruction 1: Describe how to transition from the reference start image to the reference end image to generate Image B, including explicit intermediate transition path constraints.
- Instruction 2: Describe how to transition from the reference start image to the reference end image to generate Image C, including explicit intermediate transition path constraints.
- Image A: Composed of two parts—left/top as the reference start image, right/bottom as the reference end image.
- Image B: The intermediate transition process result generated from Image A's start/end images (should comply with the path constraints in Instruction 1).
- Image C: The intermediate transition process result generated from Image A's start/end images (should comply with the path constraints in Instruction 2).

Evaluation Task:
Determine whether the model truly understands and clearly expresses the intermediate transition process from start to end, strictly follows the path constraints in Instruction 1 and Instruction 2 respectively, and reflects differentiation between the two paths.

Scoring Criteria (Maximum 5 points):
Do not relax the standard due to overall stylistic harmony or subjective speculation of intent; score only based on explicitly presented content in Images B and C. Please do not assign a higher score simply because the overall style appears coordinated or reasonable. Use the strictest and most conservative standard for judgment.
- 5 points (Complete Understanding): Both B and C accurately, clearly, and with high quality reproduce the full transition process, strictly conforming to their respective path constraints; demonstrates strong understanding and differentiation ability.
- 4 points (Good Understanding): B and C reflect the transition process well, meet the corresponding path constraints, and show generally good understanding.
- 3 points (Average Understanding): B and C roughly present the transition process, basically reflect the path constraints, but contain inaccuracies.
- 2 points (Poor Understanding): B and C show transitions but lack clear path differentiation or fail to fully implement the constraints; unable to generate according to the required paths.
- 1 point (No Understanding): B and C cannot reasonably reflect the intermediate process, paths are invalid/chaotic, do not match the textual instructions.

Key Evaluation Points (Check item by item):
- Explicitness and completeness of intermediate process: (1) Do B and C clearly show "intermediate steps," rather than simply copying or slightly modifying the start/end states? (2) Steps must be presented sequentially in a grid format (each grid as one stage, with the stage number in the top-left corner); do not rely on common sense or assumed knowledge to fill in unexpressed steps.
- Conformance to path constraints (verify item by item): (1) In B and C, does each step explicitly correspond to the path constraints described in their respective instructions (explicit evidence only)? (2) "Looks reasonable overall" cannot substitute for explicit compliance.
- Path understanding and differentiation ability: (1) Under different path constraints, do B and C show distinct intermediate processes and stage sequences? (2) Check for skipped stages, redundant stages, or missing stages, and deduct points accordingly.

Examples:
- "B explicitly shows the intermediate process path, but deviates somewhat from the path requirements; C's final result fits, but intermediate steps contain stage skipping/redundancy, failing to reflect the complete path process."
- "Final score: 2"

Input:
- Instruction 1: {Instruction_A}
- Instruction 2: {Instruction_B}
- Image A: First uploaded photo.
- Image B: Second uploaded photo.
- Image C: Third uploaded photo.

Output format:
After completing the evaluation, please output the result in the following format:

Final score: X

Figure 32: Prompt for evaluating Process Plausibility.

