# OpenReview forum: "InEdit-Bench: Benchmarking Intermediate Logical Pathways for Intelligent Image Editing Models"
_ICLR.cc/2026/Conference — ICLR 2026 Conference Withdrawn Submission_

### Official Review · Reviewer_fAh1 · 2025-10-28

**Soundness:** 1
**Presentation:** 2
**Contribution:** 3
**Rating:** 4
**Confidence:** 3

**Summary:**

The paper introduces **InEdit-Bench**, a benchmark for evaluating image editing models on their ability to produce *intermediate logical pathways* between input and target images. The benchmark comprises 237 annotated instances divided into 4 categories and 16 sub-tasks (state transition, dynamic process, temporal sequence, scientific simulation). It assesses both visual quality and logical coherence using six metrics: three existing image metrics and three newly proposed process-level metrics derived from large vision-language model (VLM) judgments (GPT-4o). Fourteen models are evaluated across the benchmark, revealing that current image editing systems perform poorly on multi-step reasoning and process coherence.

**Strengths:**

- **Novel evaluation perspective:** Focuses on the reasoning process of image editing rather than only final output quality — a relevant and underexplored dimension.
- **Dataset diversity:** Includes varied categories (scientific, temporal, dynamic), providing a potentially useful testbed for assessing sequential or reasoning-based editing.
- **Broad model coverage:** Evaluates many prominent image editing models, both open-source and proprietary.

**Weaknesses:**

1. **Ad-hoc task taxonomy.** The division into four categories and sixteen sub-tasks lacks principled justification. Definitions between task types (e.g., “temporal sequence” vs “dynamic process”) are blurry.
2. **Unvalidated evaluation method.** The three novel metrics depend entirely on GPT-4o prompts with no human validation or correlation study. Human evaluation is essential to confirm alignment of VLM scores with human judgment.
3. **Undefined “accuracy.”** Table 1 reports “Accuracy” without defining how it is computed for multi-stage edits.
4. **Limited and uninformative analysis.** The result discussion is descriptive but yields few insights into why models fail or what differentiates task difficulty.
5. **Small dataset.** Only 237 instances are used, which makes it hard to justify the lack of human evaluation.

**Questions:**

1. How exactly is *accuracy* computed in Table 1? Please provide a formal definition.
2. Have you performed any **human evaluation** to validate that GPT-4o judgments align with human perception of logical or scientific plausibility? If not, can you report at least a small-scale correlation study?
3. Would you consider expanding the benchmark or reporting confidence intervals to strengthen statistical reliability?

---

### Official Review · Reviewer_sRaq · 2025-10-29

**Soundness:** 2
**Presentation:** 2
**Contribution:** 1
**Rating:** 2
**Confidence:** 3

**Summary:**

This paper proposes a new benchmark for image editing tasks. It focuses on a new task: building a realistic pathway between two images. This allows to evaluate the capabilities of a model to generate realistic and physically plausible transformations from source to target image. The authors propose a new dataset of pairs (image source, image target), and an evaluation methodology based on pre-trained VLMs. They then evaluate several large text-to-image generative models on their proposed benchmarks.

**Strengths:**

* Propose a novel benchmark for image editing with new data and new evaluation methodology.

* Propose a new task of evaluating generation quality along editing path.

**Weaknesses:**

* This benchmark heavily relies on AI for both dataset construction and evaluation metrics. While this can be seen as a strength (automatization), the authours should validate the AI components. First, the generated images seem sometimes of poor quality. For example, I am concerned over the soundness of the "Science" part of the benchmark. Moreover, the evaluation methodology also relies on VLMs. This raises questions about robustness and correlation with human perception of the used VLM. How strong are the evaluation capabilities of the model used? What are the failure cases of the VLM as a judge? These are crucial questions that should be addressed to ensure the benchmark is valid and reliable.

* The dataset is partly composed of generated images. This leads to several problems. 1) Some of the generated images seem to be poor of poor quality. 2) There is a bias for model evaluation. Indeed, the best model according to the benchmark is GPT-Image-1. However, it is also the model that has been used for constructing the benchmark. But these two facts can be correlated: it could be easier for a model to work on images that it has generated itself.

* There lacks details about the benchmark construction.

**Questions:**

See weaknesses

---

### Official Review · Reviewer_P9Pp · 2025-11-01

**Soundness:** 3
**Presentation:** 3
**Contribution:** 2
**Rating:** 2
**Confidence:** 3

**Summary:**

The paper introduces InEdit-Bench, a benchmark for evaluating whether image editing models can generate intermediate, causally consistent visual steps between a source and target image. It covers 4 categories / 16 tasks and uses an LMM-as-judge to score both visual quality and process plausibility across multi-grid outputs.

**Strengths:**

- Well-motivated task: moves evaluation from “final image only” to “full editing trajectory,” which is missing in current benchmarks.

- Process-oriented metrics: adding logical/scientific plausibility on top of standard vision metrics is a concrete contribution.

**Weaknesses:**

- Small scale (237 cases) for a benchmark that aims to compare many models; unclear robustness.

- Single judge dependency: relies heavily on one LMM evaluator; no human–LMM agreement or judge ablations.

**Questions:**

- How do you validate that the LMM-as-a-judge scores (especially for logical coherence and process plausibility) actually correlate with human judgments across different task types in InEdit-Bench?

- In many of the paper’s examples, the images have complex compositions, which makes evaluation harder. The differences between images in an image sequence may be very small, which can also make evaluation quite challenging. Why did you choose GPT-4o instead of a newer, more capable model?

- The dataset is quite small but covers many categories. Why did you design only 237 examples? Did you consider fully synthetic or semi-automatic pipelines to generate more high-quality data?

- How sensitive are results to the number/layout of grids?

---

### Official Review · Reviewer_bjzx · 2025-11-03

**Soundness:** 2
**Presentation:** 2
**Contribution:** 1
**Rating:** 2
**Confidence:** 4

**Summary:**

The paper proposes InEdit-Bench, a benchmark for multi-step, reasoning-aware image editing. It curates 4 categories (state transition, dynamic process, temporal sequence, scientific simulation) spanning 16 sub-tasks, and introduces 6 automatic metrics to score “intermediate logical pathways.” The paper reports results for 14 editing models.

**Strengths:**

Interesting problem: evaluation beyond single-shot edits.

Attempt to structure multi-step edits with sub-tasks.

**Weaknesses:**

I have several issues with this paper, and not sure where should I start. But here is my attempt.

**Major Conceptual Issues**

1. Fundamental Mischaracterization of the Task.

My biggest concern is that the paper conflates visual interpolation/transition generation with reasoning. Generating intermediate frames between two images is primarily an interpolation or perhaps an animation task, not a reasoning task. The claim that this measures "procedural reasoning" and "causal understanding" is exaggerated. Models are simply being asked to fill in plausible visual transitions, which doesn't require deep reasoning about causality.

2. Artificial and Problematic Task Design.

The requirement to generate a single image divided into N grids is highly artificial and not representative of real-world use cases. This design choice introduces several problems: it creates grid artifacts that evaluators are instructed to ignore (undermining evaluation validity), forces an unnatural output format that no existing model was designed for, and makes the task unnecessarily difficult in ways unrelated to actual reasoning ability

**Evaluation Methodology Flaws**

3. Circular Evaluation Logic

Using GPT-4o as the evaluator while simultaneously evaluating GPT-based models creates a conflict of interest. GPT-4V may be biased toward outputs similar to what GPT models produce, and hence, perhaps the reason for its consistently outperforming other models quite often.

4. Insufficient Scale

Only 237 instances across 16 subtasks are too small for a benchmark paper.

5. No Human Validation

The paper relies entirely on automated evaluation (GPT-4o) without any human studies to validate that the metrics actually measure what they claim. LLM-as-a-judge sounds cute, but it is unreliable and accelerates AI slop. The core task here is reasoning-aware, multi-step editing, which only matters if humans vet the intermediate generations and the causal chain. Otherwise, we get a loop where Model A produces outputs, Model B “judges” them, and we cannot tell if we made scientific progress.

6. Questionable Metric Validity

The three *novel* metrics lack proper validation. There is no ablation studies showing they measure distinct capabilities. There is no evidence that high scores correlate with actual reasoning ability.

**Experimental Design Problems**

7. Unfair Comparison Setup.

Open-source models receive concatenated images with black-and-white striped lines, which could confuse models not trained on such inputs.

8. Missing Critical Baselines

The paper doesn't evaluate video generation models, which are explicitly designed for temporal/sequential generation. This is a glaring omission since video models directly address the paper's stated goal and should be the primary model used for this evaluation.



**Methodological Issues**


9. Grid Line Artifacts Should Be Ignored

The instruction to ignore artifacts introduced by the grid format (Section 3.3) is methodologically unsound. If the format introduces systematic artifacts, the benchmark design is flawed, not the model outputs.

10. Lack of Error Analysis

No qualitative analysis of failure modes or discussion of why models fail. Are failures due to: a) misunderstanding of instructions? b) Inability to generate grids? c) Actual reasoning deficits? d) Ambiguity in the LLM-as-judge?

**Limited Practical Value**

11. Unclear Real-World Application

The paper doesn't establish why generating intermediate pathways in a grid format is useful for practical image editing applications. Most real editing scenarios don't require explicit multi-step visualization.

**Questions:**

1) How do scores change if you swap the judge with, say, Gemini?

2) Given the sequential/temporal nature of the tasks, why weren’t state-of-the-art video generation/editing models included as primary baselines?

3) Why were humans not used to calibrate the automatic metrics?

4) Can you quantify what fraction of failures arises from: (a) instruction misunderstanding, (b) inability to produce the N-grid format, (c) genuine reasoning deficits, (d) judge ambiguity?

---

### Note · Authors · 2025-11-14

I have read and agree with the venue's withdrawal policy on behalf of myself and my co-authors.